# Harnessing Model Uncertainty for Adaptive Causal Debiasing in Visual Question Answering

## Abstract

Visual Question Answering (VQA) models often exploit spurious correlations, hindering true multimodal reasoning. While causal inference offers principled debiasing methods, current approaches pair complex causal graphs with overly simplistic, static counterfactual interventions (e.g., feature subtraction). This limits effectiveness. We challenge this by proposing a novel framework synergistically integrating uncertainty estimation with causal counterfactual reasoning for robust VQA debiasing. This is the first work, to our knowledge, to leverage uncertainty within a causal VQA framework. We systematically explore uncertainty quantification techniques (entropy, prediction margin) to assess model confidence. This estimated uncertainty dynamically modulates the counterfactual intervention, allowing adaptive adjustment of biased information sources based on real-time confidence. This moves beyond rigid interventions. Furthermore, we introduce a tailored Curriculum Learning strategy that dynamically assesses sample difficulty using uncertainty-aware metrics, enhancing the adaptive mechanism. Our uncertainty-guided intervention module is architecture-agnostic, enabling integration into diverse VQA networks. This adaptive, uncertainty-aware approach offers a more flexible, robust, and theoretically grounded pathway towards mitigating VQA biases.

## 1 Introduction

Visual Question Answering (VQA) stands as a benchmark task in multimodal artificial intelligence, challenging models to reason jointly over visual and textual information to answer questions about images Antol et al. (2015). While significant progress has been made, state-of-the-art VQA models often exhibit undesirable behaviors, relying heavily on linguistic priors and spurious correlations present in datasets rather than performing robust, grounded reasoning. This tendency leads to models that perform well on in-distribution test sets but fail to generalize to real-world scenarios or answer questions requiring genuine compositional understanding Agrawal et al. (2016); Goyal et al. (2017).

Mitigating these biases is crucial for building trustworthy and capable VQA systems. Causal inference has emerged as a principled and powerful framework for addressing this challenge. By explicitly modeling the causal relationships between questions, images, answers, and potential confounding factors (like linguistic priors), researchers aim to identify and neutralize biasing pathways Niu et al. (2021). Methodologies often involve constructing causal graphs and applying criteria like the backdoor adjustment to guide model training or inference Song et al. (2024); Liu et al. (2024); Pan et al. (2024); Vosoughi et al. (2024); Nguyen & Okazaki (2023); Patil et al. (2023).

However, a critical limitation persists within many current causal VQA debiasing approaches. While significant effort is invested in formulating sophisticated causal graphs to diagnose the sources of bias, (a comprehensive summary of contemporary methodologies utilizing such complex causal graph analyses is deferred to Appendix), the mechanisms used to perform counterfactual interventions during training or inference often remain surprisingly simplistic and static. Many methods resort to rudimentary operations like directly subtracting or zeroing-out biased feature representations, implicitly pushing the biased pathway toward a non-informative state (e.g., a uniform predictive distribution). However, this fixed, "one-size-fits-all" correction is not optimal: it fails to account for

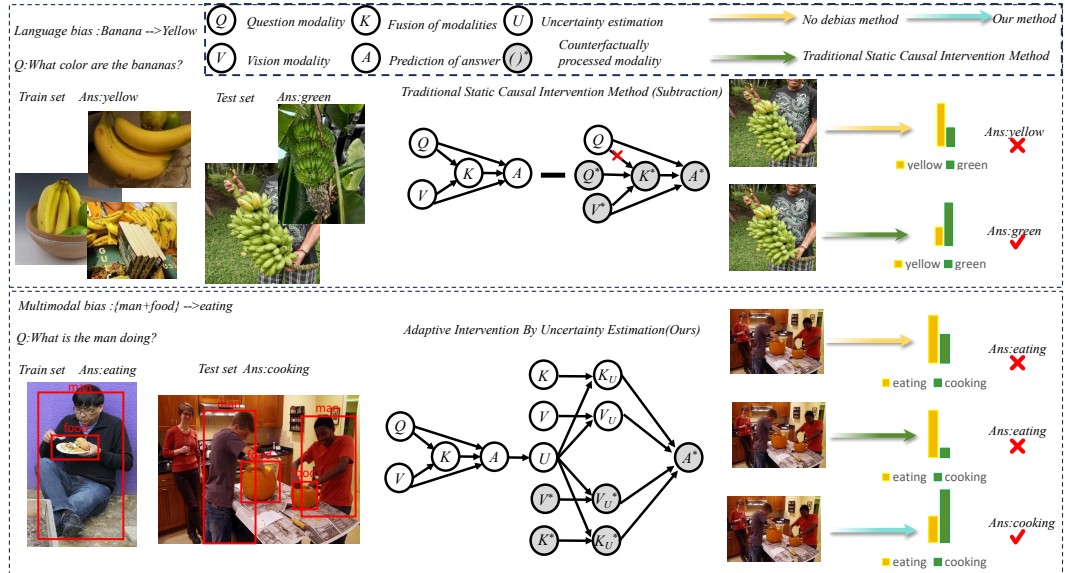

Figure 1: Our uncertainty-guided approach versus traditional static debiasing. **Top:** A static causal intervention, while sometimes effective, applies a fixed correction for a language bias ("Banana $\rightarrow$ Yellow"). **Bottom:** Our method introduces an adaptive intervention for a multimodal bias ("man+food $\rightarrow$ eating"). It uses uncertainty ($U$) to dynamically modulate how factual ($K, V$) and counterfactual ($K^*, V^*$) information is integrated, leading to a more robust final prediction.

the varying degrees of bias present across different data instances. They apply a uniform intervention regardless of whether the model is confident or uncertain about its internal representations, potentially over-correcting in some cases and under-correcting in others, limiting the overall effectiveness of the debiasing process.

In this work, we argue for a more nuanced and adaptive approach to causal intervention in VQA (illustrated in Figure 1). We propose to bridge the gap between causal theory and practical intervention mechanisms by leveraging model uncertainty. Our core hypothesis is that a model's internal uncertainty about its predictions or representations can serve as a valuable, dynamic signal to guide the strength and nature of causal interventions. Intuitively, higher uncertainty might indicate a greater reliance on potentially biased shortcuts, suggesting a stronger intervention is needed, while lower uncertainty might warrant a more subtle adjustment. To the best of our knowledge, this paper presents the first integration of uncertainty estimation techniques within a causal counterfactual reasoning framework specifically for VQA debiasing. We demonstrate that uncertainty estimation is not only compatible with but also highly complementary to causal debiasing goals.

To realize this, we introduce an Adaptive Uncertainty-Guided Intervention module. This module dynamically estimates the model's uncertainty during processing, employing and systematically exploring various well-established uncertainty quantification techniques such as prediction entropy, confidence margin, or variations from ensemble methods Hendrycks & Gimpel (2017); Lakshminarayanan et al. (2017); Carlini & Wagner (2017). The estimated uncertainty then directly modulates the counterfactual intervention process, allowing the model to adaptively adjust the influence of different information sources (e.g., potentially biased language features vs. visual evidence) on a per-instance basis. This adaptive mechanism replaces the rigid, one-size-fits-all interventions common in prior work.

Furthermore, recognizing that the learning process itself can benefit from adaptivity, we introduce a novel Curriculum Learning (CL) strategy specifically designed to synergize with our uncertainty-aware approach. Distinct from previous CL applications in VQA that often rely on predefined or static difficulty metrics Pan et al. (2022), our strategy dynamically assesses sample difficulty using uncertainty-aware metrics derived directly from the model's state or preliminary evaluations. This allows for a training progression with strong logical coherence: the model first learns from

less ambiguous examples where it is more confident, gradually moving towards more challenging instances where its uncertainty (and thus the adaptive intervention mechanism) plays a more critical role. This uncertainty-driven curriculum aligns perfectly with our core technical contribution.

Crucially, our proposed uncertainty-guided intervention module is designed with modularity in mind, making it architecture-agnostic. It can be readily incorporated into various existing VQA network backbones without requiring fundamental architectural redesigns, facilitating broader adoption and experimentation.

In summary, our main contributions are: (1) The first framework to synergistically integrate **uncertainty estimation with causal counterfactual reasoning** for VQA debiasing. (2) An **adaptive intervention mechanism**, which includes the systematic exploration and application of multiple uncertainty estimation techniques, that **dynamically modulates causal adjustments** based on quantified model uncertainty, moving beyond static intervention approaches. (3) A novel, **uncertainty-aware Curriculum Learning strategy** that aligns training difficulty with the model's adaptive capabilities. (4) A modular, **architecture-agnostic design** ensuring broad applicability across different VQA models.

We demonstrate through extensive experiments that our adaptive, uncertainty-guided approach significantly improves debiasing performance and robustness compared to baseline methods. The rest of the paper is organized as follows: Section 2 discusses related work. Section 3 details our proposed methodology. Section 4 presents the experimental setup and results. Finally, Section 5 concludes the paper.

## 2 Related Work

This section reviews prior work in VQA debiasing, causal inference for VQA, uncertainty estimation, and curriculum learning, contextualizing our approach.

### 2.1 Bias in Visual Question Answering

Dataset bias in VQA, especially strong language priors, is widely recognized. Models often answer based on question text, ignoring visual cues, leading to poor generalization and lacking true multimodal understanding Agrawal et al. (2018); Ma et al. (2024). Early efforts to mitigate these included dataset manipulation Chen et al. (2023); Gupta et al. (2022); Chen et al. (2022), specialized contrastive learning for modality balancing Si et al. (2022a); Liu et al. (2023b); Hao et al. (2024), and ensemble or adversarial training Cadene et al. (2019b); Han et al. (2023); Liang et al. (2021). While beneficial, these often lack a principled framework for addressing the causal mechanisms of bias.

### 2.2 Causal Inference for VQA Debiasing

Causal inference offers a structured approach by modeling cause-and-effect relationships. Several works apply causal principles to VQA debiasing: A prominent line leverages counterfactual queries—asking "what if the input differed?" The CF-VQA framework Niu et al. (2021) generates counterfactual samples (e.g., by masking inputs) and enforces consistency between factual/counterfactual predictions to disentangle spurious correlations. Building on this, subsequent research explored more sophisticated counterfactual generation and application, such as for knowledge distillation targets Pan et al. (2022), dual-debiasing for specific linguistic biases Song et al. (2024), or "possible worlds" reasoning for complex confounds Vosoughi et al. (2024).

Despite their theoretical appeal, these causal approaches, particularly with counterfactuals, often rely on **static, simplistic interventions**. For example, methods like CF-VQA might subtract feature vectors or zero-out attention weights. Such fixed interventions lack adaptability to varying model confidence or sample characteristics. This rigidity can limit effectiveness against diverse bias manifestations, a limitation our work addresses with uncertainty-guided adaptive intervention.

### 2.3 Uncertainty Estimation in Deep Learning

Model uncertainty estimation is crucial for reliability, OOD detection, and active learning. Techniques include: Bayesian Neural Networks (BNNs) that place distributions over weights He (2023)

(computationally expensive); Ensemble Methods using prediction variance from multiple models Lakshminarayanan et al. (2017) (effective but costly); and Deterministic and Other Methods like predictive entropy, prediction margin, or evidential deep learning Hendrycks & Gimpel (2017); Carlini & Wagner (2017).

While widely used in vision and NLP Kahl et al. (2024); Zha et al. (2024), the role of uncertainty in VQA debiasing is evolving. Prior works like LBSD Yuan et al. (2022) have leveraged it to guide knowledge distillation, using uncertainty to shape the training objective. Our work proposes a fundamentally different approach. We are the first, to our knowledge, to use uncertainty to **dynamically guide a causal intervention**.

### 2.4 CURRICULUM LEARNING

Curriculum Learning (CL) trains models on examples in a meaningful, typically easy-to-hard, order, potentially improving convergence and generalization Bengio et al. (2009). In VQA, prior CL often used predefined heuristics (e.g., question length, answer frequency) or initial model loss for difficulty Aissa et al. (2023); Nguyen et al. (2025). Some works used CL for debiasing by gradually introducing harder, more biased examples Pan et al. (2022); Zheng et al. (2024).

However, existing VQA CL often uses static or heuristic difficulty measures. Our work introduces a CL strategy where the curriculum is **dynamically informed by uncertainty-aware metrics**. This couples the training regimen with our uncertainty-guided intervention, ensuring the model progressively learns to handle samples needing its adaptive capabilities, offering a more principled curriculum than prior methods.

## 3 METHODOLOGY: ADAPTIVE UNCERTAINTY-GUIDED COUNTERFACTUAL INTERVENTION

Our methodology refines foundational counterfactual debiasing principles and introduces our core contributions: the Adaptive Uncertainty-Guided Intervention (AUGI) module and an Uncertainty-Aware Curriculum Learning (UACL) strategy. These components work synergistically to achieve robust and adaptive VQA debiasing. Figure 2 provides an overview.

### 3.1 BACKGROUND: BASELINE VQA ARCHITECTURE

Our framework intervenes on a baseline causal model built upon an UpDn-style VQA architecture Anderson et al. (2018); Niu et al. (2021). The backbone first encodes the image as a set of region-level visual features $V = \{v_i\}$ and the question as a sequence representation summarized by a hidden state $q$. Following Niu et al. (2021), these representations are processed through three parallel branches designed to isolate specific causal pathways: (1) a question-only pathway, modeling $Q \to A$, which maps $q$ to a question feature $Z_Q$; (2) a vision-only pathway, modeling $V \to A$, which maps $V$ to a visual feature $Z_V$; and (3) a multimodal pathway, modeling $V, Q \to K \to A$, which uses attention over $V$ conditioned on $q$ to produce a joint representation $Z_{VQ}$. The three pathway features are then fused by a function $F$ (e.g., element-wise summation) to produce the Total Effect logit $z_{TE} = F(Z_{VQ}, Z_Q, Z_V)$. Auxiliary losses on the $Z_Q$ and $Z_V$ branches are used during training to ensure each pathway is meaningful. This baseline causal architecture provides the foundation for the counterfactual interventions we describe next.

### 3.2 FOUNDATIONAL COUNTERFACTUAL DEBIASING IN VQA

Visual Question Answering models ($\text{Model}(\cdot, \cdot; \theta)$) typically map an image $V$ and question $Q$ to answer probabilities $p_{ans} = \sigma(\text{Model}(V, Q; \theta))$. Counterfactual debiasing methods, like the CF-VQA baseline Niu et al. (2021) our work builds upon, aim to mitigate biases (e.g., linguistic shortcuts) by dissecting the prediction process. Such models often fuse information from joint visual-linguistic ($Z_{VQ}$), question-only ($Z_Q$), and vision-only ($Z_V$) pathways using a fusion function $F(\cdot, \cdot, \cdot)$.

The core idea involves comparing a **Total Effect (TE)** logit, $z_{TE} = F(Z_{VQ}, Z_Q, Z_V)$, with a counterfactually derived **Natural Direct Effect of the Question (NDE$_Q$)**. The $z_{NDE_Q}$ is computed by isolating the question's influence, typically by replacing $Z_{VQ}$ and $Z_V$ with a non-informative

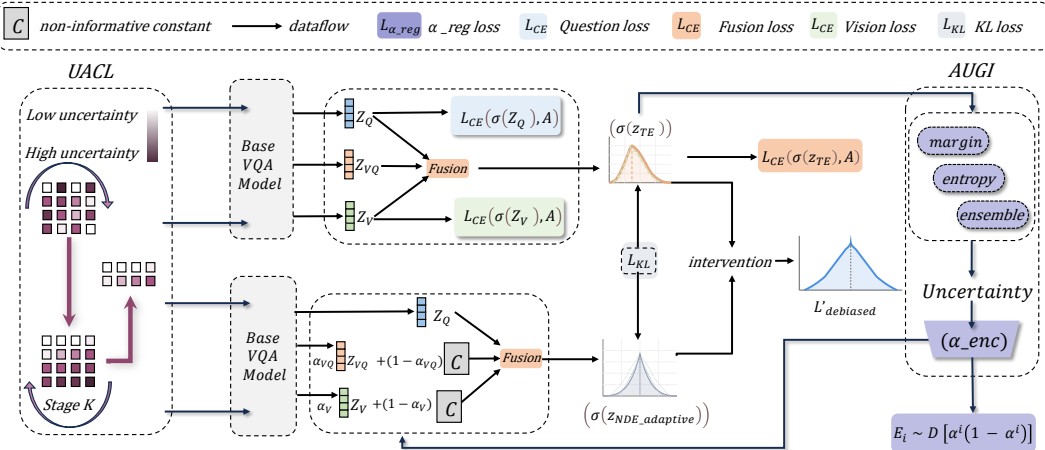

Figure 2: This figure (detailed in Sec. 3) presents our adaptive VQA debiasing architecture, where Uncertainty-Aware Curriculum Learning (UACL) and Adaptive Uncertainty-Guided Intervention (AUGI) operate synergistically. UACL (left) employs initial uncertainty to stage training data ("Stage K"). The core model processes factual inputs (V,Q) to yield $Z_{TE}$. AUGI (right) then leverages runtime uncertainty ($U$)—derived from "margin", "entropy", or "ensemble"—to compute an adaptive factor $\alpha$. This $\alpha$ dynamically modulates the counterfactual pathway: $Z_{VQ}$ and $Z_V$ are interpolated with a constant $C$ to form $z_{NDE\_adaptive}$.

constant $c$ within the fusion: $z_{NDE_Q} = F(c, Z_Q, c)$. The debiased prediction $L_{debiased}$ is then obtained by subtraction: $L_{debiased} = z_{TE} - z_{NDE_Q}$.

To ensure the stability and semantic relevance of this subtraction, a Kullback-Leibler (KL) divergence loss, $L_{KL} = D_{KL}(\sigma(z_{TE})||\sigma(z'_{NDE_Q}))$, is often employed. Here, $z'_{NDE_Q}$ uses detached inputs for stability. This $L_{KL}$ encourages the question-only pathway (producing $Z_Q$) to generate a distribution similar to the total effect, making $z_{NDE_Q}$ a more reliable proxy for the question's (potentially biased) contribution, thereby making its subtraction more effective for debiasing. The primary VQA loss, $L_{CE\_TE}$, is typically applied to $z_{TE}$.

## 3.3 ADAPTIVE UNCERTAINTY-GUIDED COUNTERFACTUAL INTERVENTION (AUGI)

The static intervention described above applies a uniform correction. AUGI introduces adaptivity by dynamically modulating this intervention based on model uncertainty.

**Uncertainty Quantification:** We first estimate model uncertainty $U$ for an instance, which can be derived from $z_{TE}$, or intermediate outputs $Z_Q, Z_V$. Let $p_{ans} = \sigma(z_{TE})$ be the predictive probability distribution over $C$ answers. We explore several UncertaintyFunc($p_{ans}$):

- **Predictive Entropy** ($U_H$): Captures overall predictive dispersion.

$$U_H(p_{ans}) = -\sum_{j=1}^{C} p_{ans,j} \log p_{ans,j} \tag{1}$$

- **Prediction Margin** ($U_{Margin}$): Highlights confusion between the top choices. Let $p_{top1}$ and $p_{top2}$ be the highest and second-highest probabilities in $p_{ans}$.

$$U_{Margin}(p_{ans}) = 1 - (p_{top1} - p_{top2}) \tag{2}$$

- **Ensemble Disagreement** ($U_{Ens}$): Utilizes variance among predictions from $M$ model heads $\{p_{ans}^{(m)}\}_{m=1}^{M}$, e.g., $U_{Ens} = \frac{1}{C}\sum_{j=1}^{C}\text{Var}_m(p_{ans,j}^{(m)})$, reflecting model stability.

These metrics offer diverse signals. The raw uncertainty score $U$ (e.g., $U_H^{(i)}, U_{Margin}^{(i)}, U_{Ens}^{(i)}$ for a sample $i$) is first normalized to ensure stable input to the transformation function. We use standard

score normalization:

$$U_{norm}^{(i)} = \frac{U^{(i)} - \mathbb{E}[U]}{\sqrt{\text{Var}[U] + \epsilon_{norm}}} \tag{3}$$

where $\mathbb{E}[U]$ and $\text{Var}[U]$ can be batch statistics or running averages, and $\epsilon_{norm}$ is a small constant for numerical stability. This normalized score $U_{norm}^{(i)}$ is then mapped to an adaptive modulation factor $\alpha_p^{(i)}$ for each pathway $p \in \{VQ, V\}$. This is achieved using a learnable scaled sigmoid function whose parameters $(\gamma_p, \delta_p)$ are unique to each pathway:

$$\alpha_p^{(i)} = f_{U,p}(U_{norm}^{(i)}) = \frac{1}{1 + \exp\left(-\gamma_p \cdot \left(U_{norm}^{(i)} - \delta_p\right)\right)}, \quad p \in \{VQ, V\} \tag{4}$$

where the parameters $\gamma_p$ (steepness) and $\delta_p$ (center point) are learned independently for each pathway, dynamically shaping the responsiveness of each $\alpha_p$ to uncertainty. To ensure that higher normalized uncertainty $U_{norm}$ leads to a larger intervention factor (i.e., a monotonically increasing mapping), we enforce $\gamma_p > 0$ by parameterizing it as $\gamma_p = \text{softplus}(\tilde{\gamma}_p)$ with an unconstrained underlying parameter $\tilde{\gamma}_p$.

**Adaptive Intervention Mechanism:** Instead of fully replacing $Z_{VQ}$ and $Z_V$ with $c$, AUGI uses the respective pathway-specific alpha factors, $\alpha_{VQ}^{(i)}$ and $\alpha_V^{(i)}$, to interpolate, yielding intervened features $Z'_{VQ}$ and $Z'_V$:

$$Z'_{VQ} = \alpha_{VQ}^{(i)} \cdot Z_{VQ} + (1 - \alpha_{VQ}^{(i)}) \cdot c \tag{5}$$

$$Z'_V = \alpha_V^{(i)} \cdot Z_V + (1 - \alpha_V^{(i)}) \cdot c \tag{6}$$

The question pathway $Z_Q$ remains unaltered here, as $z_{NDE\_adaptive}$ aims to quantify the adaptively estimated full effect of the question, where adaptation arises from how $Z_{VQ}$ and $Z_V$ contextualize it. The adaptively formulated Natural Direct Effect $z_{NDE\_adaptive}$ is then:

$$z_{NDE\_adaptive} = F(Z'_{VQ}, Z_Q, Z'_V) \tag{7}$$

This allows each $\alpha$ factor to dynamically control the intervention's severity. Here, $\alpha_p$ gates the influence of the original feature $Z_p$. When uncertainty is high, $\alpha_p$ approaches 1. This causes the counterfactual term $z_{NDE\_adaptive}$ to become very similar to the total effect $z_{TE}$, resulting in a strong intervention where the final debiased logit $L'_{debiased}$ is pushed towards zero. Conversely, when uncertainty is low, $\alpha_p$ approaches 0. This makes $z_{NDE\_adaptive}$ rely on the non-informative constant $c$, resulting in a weaker, baseline intervention. This creates a nuanced spectrum of debiasing strength, directly responsive to instance-specific ambiguity—a significant advance over fixed strategies.

**AUGI Debiased Prediction and Regularization:** The final debiased prediction is $L'_{debiased} = z_{TE} - z_{NDE\_adaptive}$. The $L_{KL}$ (from Sec. 3.2) remains vital for robustly training $Z_Q$. To prevent the adaptive mechanism from collapsing to trivial states (i.e., $\alpha$ values always being 0 or 1), we introduce a regularization term that encourages them to take values within the $(0, 1)$ range. This regularization is formulated as the sum of expectations of variance-like terms for each $\alpha$:

$$L_{\alpha\_reg} = -\lambda_{reg} \cdot \mathbb{E}_{i \sim \mathcal{D}}[\alpha_{VQ}^{(i)}(1 - \alpha_{VQ}^{(i)}) + \alpha_V^{(i)}(1 - \alpha_V^{(i)})] \tag{8}$$

where $\lambda_{reg}$ is a hyperparameter controlling the strength of this regularization, and the expectation is taken over the data samples $i$.

### 3.4 UNCERTAINTY-AWARE CURRICULUM LEARNING (UACL)

UACL optimizes training by sequencing samples from easy to hard, with difficulty defined by model uncertainty, creating synergy with AUGI.

**Initial Difficulty Assessment:** At the outset of UACL, an initial difficulty score $U_i^{init}$ is computed for each training sample $(V_i, Q_i, A_i) \in \mathcal{D}$ based on the model's initial state $\theta_{init}$ (e.g., after a few warm-up epochs or from a pre-trained model):

$$U_i^{init} = \text{UncertaintyFunc}(\sigma(\text{Model}(V_i, Q_i; \theta_{init}))) \quad \forall i \in \{1, \ldots, N\} \tag{9}$$

This initial uncertainty serves as a proxy for sample difficulty, leveraging the model's early learning dynamics where it typically shows lower uncertainty for inherently simpler instances.

**Curriculum Pacing and Subset Selection:** Training proceeds through $K$ stages. At each curriculum stage $k \in \{1, \ldots, K\}$, the model is trained on a progressively larger and more difficult subset of the data, $\mathcal{D}_k$. This subset is defined by including all samples whose initial uncertainty $U_i^{init}$ falls below the current difficulty threshold $T_k$ (derived from Equation 11):

$$\mathcal{D}_k = \{(V_i, Q_i, A_i) \in \mathcal{D} \mid U_i^{init} \leq T_k(k, K, \mathbb{E}[U^{init}], \text{Var}[U^{init}], p)\} \tag{10}$$

where $T_k(\cdot)$ is the pacing function that dynamically expands the training set by considering stage $k$, total stages $K$, overall statistics of initial uncertainties (e.g., mean $\mathbb{E}[U^{init}]$ and variance $\text{Var}[U^{init}]$ to set $T_{min}, T_{max}$ bounds), and the pacing exponent $p$. The pacing function itself is defined as:

$$T_k = T_{min} + (T_{max} - T_{min}) \cdot \left(\frac{k}{K}\right)^p \tag{11}$$

where $T_{min}$ and $T_{max}$ can be set, for example, based on percentiles of the $U_i^{init}$ distribution. This gradual exposure to harder samples (higher $U_i^{init}$) prevents overwhelming the model initially and allows it to build a robust foundation. Crucially, as the model sees progressively more challenging instances, the AUGI mechanism, refined on simpler data, is better prepared to apply effective adaptive interventions. This dynamic, uncertainty-driven scheduling distinguishes UACL from prior VQA CL methods relying on static heuristics (e.g., question length), by intrinsically linking the curriculum to the model's evolving confidence and its adaptive debiasing capacity.

### 3.5 FINAL TRAINING OBJECTIVE AND OPTIMIZATION

The overall objective at curriculum stage $k$ for parameters $\theta$ is:

$$L_{Core}^{(k)} = L_{CE\_TE}(\sigma(z_{TE}), A) + \lambda_{KL} L_{KL} \tag{12}$$

$$L_{Total}^{(k)}(\theta) = \mathbb{E}_{(V,Q,A) \in \mathcal{D}_k}\left[L_{Core}^{(k)} + \sum_{path \in \{Q,V\}} \lambda_{path} L_{CE}(\sigma(Z'_{path}), A) + L_{\alpha\_reg}(\alpha)\right] \tag{13}$$

where $\lambda$ terms are hyperparameters, and $Z'_Q, Z'_V$ are outputs from the question and vision pathways respectively. Optimization uses standard techniques (e.g., Adam), and inference uses $L'_{debiased}$. A sensitivity analysis of the key hyperparameters is provided in Appendix B.3 (Table 3).

## 4 EXPERIMENTS

### 4.1 EXPERIMENTAL SETUP

We evaluate our approach primarily on VQA-CP v2 Agrawal et al. (2018), a dataset explicitly designed to measure robustness against language bias, and VQA v2 Goyal et al. (2017) to assess performance on the standard distribution. Evaluation follows standard VQA metrics (overall accuracy; detailed per-type accuracy for VQA-CP v2). We compare against vanilla backbones (UpDnAnderson et al. (2018), SANYang et al. (2016), SMRLCadene et al. (2019a)) and state-of-the-art debiasing techniques, encompassing traditional (e.g., CSS Chen et al. (2023), LSP Liu et al. (2023b)) and causal methods (e.g., CF-VQA Niu et al. (2021), PWVQA Vosoughi et al. (2024)). Implementation specifics, including hyperparameter choices and backbone details, are in Appendix A, and additional experiments on transformer-based backbones (BLIP, LXMERT) further validating the architecture-agnostic nature of our framework are provided in Appendix B.4 (Table 4).

### 4.2 MAIN RESULTS: COMPARISON WITH STATE-OF-THE-ART

Table 1 benchmarks our uncertainty-guided framework against existing methods. On the challenging VQA-CP v2 dataset, our approach (OURS + UpDn) achieves a new state-of-the-art performance of **63.36%**, significantly surpassing prior best methods like LSP (61.95%) and PWVQA (60.26%). Notably, our method demonstrates substantial gains in the notoriously difficult 'Num' category, reaching **60.05%** compared to 59.13% (PWVQA + SMRL), indicating improved reasoning capabilities beyond

simple language priors. Crucially, these debiasing gains do not compromise general performance; on VQA v2, our method remains highly competitive, achieving **68.65%** overall and leading in the 'Y/N' category, demonstrating the robustness of our adaptive mechanism. Similar strong performance is observed when using other backbones like SAN and SMRL, albeit with different absolute scores reflecting the baseline capabilities. For a finer-grained diagnostic analysis across diverse shortcut types on the VQA-VS dataset, we refer the reader to Appendix B.7 (Table 7).

Table 1: Comparison with State-of-the-Art methods on VQA-CPv2 and VQAv2 datasets. All scores are reported in percentages (%). Best result in each column is in **bold**, second best is underlined.

| Model | Baseline | VQA-CPv2 (%) | | | | VQAv2 (%) | | | |
|-------|----------|------|------|------|-------|------|------|------|-------|
| | | Eval | Y/N | Num | Other | Eval | Y/N | Num | Other |
| *Methods based on traditional debiasing (e.g., ensemble learning, contrast learning)* | | | | | | | | | |
| updn | none | 39.74 | 42.27 | 11.93 | 46.05 | 63.48 | 81.18 | 42.14 | 55.66 |
| SAN | none | 24.96 | 38.35 | 11.14 | 21.74 | 52.41 | 70.06 | 39.28 | 47.84 |
| SMRL | none | 38.46 | 42.85 | 12.81 | 43.20 | 37.13 | 41.96 | 12.54 | 41.35 |
| CSS | LMH | 58.95 | 84.37 | 49.42 | 48.21 | 59.91 | 73.25 | 39.77 | 55.11 |
| MMBS | LMH | 56.44 | 76.00 | 43.77 | 49.67 | **70.85** | 88.25 | **55.67** | **61.63** |
| RMLVQA | updn | 60.41 | 89.98 | 45.96 | 48.74 | 59.99 | 76.68 | 37.54 | 53.26 |
| GGD | updn | 59.37 | 88.23 | 38.11 | 49.82 | 65.79 | 77.26 | 52.71 | 60.52 |
| LSP | updn | 61.95 | 89.50 | 52.44 | 50.12 | 65.26 | 82.38 | 44.77 | 57.67 |
| HCCL | updn | 61.48 | 89.38 | 49.09 | 50.26 | * | * | * | * |
| HCCL | SAN | 46.28 | 54.57 | 18.89 | 49.45 | * | * | * | * |
| *Methods based on Causal Inference* | | | | | | | | | |
| CF-VQA | UpDn | 53.55 | **91.15** | 13.03 | 44.97 | 63.54 | 82.51 | 43.96 | 54.30 |
| CF-VQA | SMRL | 55.05 | 90.61 | 21.50 | 45.61 | 60.94 | 81.13 | 43.86 | 50.11 |
| CopVQA | CF-VQA | 57.80 | 91.10 | 41.60 | 46.40 | 57.80 | 81.40 | 43.80 | 52.40 |
| CVIV | updn | 60.08 | 88.85 | 40.77 | 50.30 | 61.93 | 80.01 | 40.28 | 53.91 |
| CiBi | CSS | 59.58 | 86.94 | 49.98 | 50.24 | 60.47 | 81.04 | 42.94 | 50.02 |
| CiBi | CF-VQA | 57.62 | 90.54 | 41.63 | 44.31 | 61.59 | 82.02 | 43.60 | 51.79 |
| CKCL(sub) | updn | 53.18 | 83.73 | 18.78 | 46.60 | 63.02 | 80.44 | 41.45 | 55.46 |
| CKCL(abs) | updn | 55.05 | 90.33 | 18.99 | 46.46 | 62.55 | 79.17 | 41.94 | 55.38 |
| PWVQA | updn | 59.06 | 88.26 | 52.89 | 45.45 | 62.63 | 81.80 | 43.90 | 53.01 |
| PWVQA | SMRL | 60.26 | 88.09 | 59.13 | 45.99 | 61.25 | 90.32 | 43.17 | 51.53 |
| OURS | updn | **63.36** | 90.78 | **60.05** | 50.20 | 68.65 | 90.50 | 53.54 | 51.01 |
| OURS | SMRL | 56.12 | 90.12 | 23.00 | 46.22 | 61.75 | 88.28 | 23.00 | 46.12 |
| OURS | SAN | 49.17 | 66.02 | 20.56 | **50.51** | 59.20 | 56.11 | 20.02 | 50.50 |

## 4.3 ABLATION STUDIES

To provide a detailed investigation of the individual and combined effects of our framework's components, we present a step-by-step ablation on VQA-CP v2 in Table 2. All models in this analysis use the UpDn backbone for a consistent comparison.

The analysis begins with two key baselines drawn from Table 1: the vanilla UpDn backbone (39.74%) and the static causal debiasing method CF-VQA (53.55%). The +13.81% improvement from the vanilla backbone to CF-VQA establishes the significant, albeit limited, benefit of a non-adaptive causal approach.

Our analysis then isolates the contribution of our **Adaptive Uncertainty-Guided Intervention (AUGI)** module. Replacing the static intervention of CF-VQA with AUGI yields substantial gains. Using Ensemble-based uncertainty, AUGI alone achieves 60.50% accuracy, a **+6.95%** improvement over the static method. This demonstrates the power of dynamically modulating the intervention based on model uncertainty, which is the core of our technical contribution.

Next, we evaluate the impact of our **Uncertainty-Aware Curriculum Learning (UACL)** strategy. When UACL is added to the AUGI framework, we see another consistent performance boost. For the best-performing Ensemble configuration, UACL adds another **+2.86%**, bringing the final accuracy to **63.36%**. This confirms that sequencing training samples based on uncertainty synergizes effectively with the adaptive intervention mechanism, allowing the model to build robust representations on simpler examples before tackling more ambiguous ones.

Collectively, these results decompose the total **+23.62%** improvement over the vanilla backbone, clearly attributing distinct, significant gains to the introduction of causal reasoning (+13.81%), the move to our adaptive intervention (+6.95%), and the addition of our uncertainty-aware curriculum (+2.86%). This validates that both AUGI and UACL are critical and complementary components of our framework.

Table 2: Detailed ablation study on VQA-CPv2, analyzing the contributions of our adaptive intervention (AUGI) and curriculum learning (UACL) components. All models use the UpDn backbone. Baseline scores are from Table 1. The "Gain" column quantifies the improvement of each configuration over its direct predecessor in the logical hierarchy (e.g., AUGI gain is relative to Static Causal, and AUGI+UACL gain is relative to AUGI alone).

| Configuration | Uncertainty Metric | UACL | Accuracy (%) | Gain (%) |
|---|---|---|---|---|
| *Baselines* | | | | |
| 1. updn | - | - | 39.74 | - |
| 2. Static Causal (CF-VQA) | - | - | 53.55 | +13.81 |
| *Contribution of Adaptive Intervention (AUGI)* | | | | |
| 3. AUGI (Ours) | Margin | - | 56.51 | +2.96 |
| 4. AUGI (Ours) | Entropy | - | 59.09 | +5.54 |
| 5. AUGI (Ours) | Ensemble | - | 60.50 | +6.95 |
| *Contribution of Curriculum Learning (UACL)* | | | | |
| 6. AUGI + UACL (Ours) | Margin | ✓ | 59.44 | +2.93 |
| 7. AUGI + UACL (Ours) | Entropy | ✓ | 61.94 | +2.85 |
| 8. AUGI + UACL (Ours) | Ensemble | ✓ | **63.36** | +2.86 |

## 4.4 ANALYSIS: CORRELATING ADAPTIVE INTERVENTION STRENGTH WITH EVOLVING TASK DIFFICULTY

To demonstrate that our adaptive intervention strength, $\alpha$, dynamically responds to task difficulty, we analyze its relationship with model accuracy on different question types from VQA-CP v2 throughout training. Our hypothesis is that more challenging question types (lower accuracy) should trigger stronger interventions (higher $\alpha$). For each question type $q$ and training epoch $e$, we record its accuracy $A(q, e)$ and the average intervention strength $\bar{\alpha}(q, e)$, calculated as:

$$\bar{\alpha}(q, e) = \frac{1}{|S_{q,e}|} \sum_{i \in S_{q,e}} \alpha_i(U_i) \tag{14}$$

where $S_{q,e}$ is the set of samples of type $q$ in epoch $e$, and $\alpha_i(U_i)$ is the intervention for sample $i$ with uncertainty $U_i$ (from Section 3.2, e.g., Eq. 4). We then compute the Pearson correlation coefficient $\rho_q$ between the accuracy trajectory $\mathbf{A}_q = [A(q, e_1), \dots, A(q, e_M)]$ and the intervention trajectory $\bar{\mathbf{A}}_q = [\bar{\alpha}(q, e_1), \dots, \bar{\alpha}(q, e_M)]$ for $M$ epochs:

$$\rho_q = \frac{\sum_{j=1}^{M}(A(q, e_j) - \mu_{\mathbf{A}q})(\bar{\alpha}(q, e_j) - \mu_{\bar{\mathbf{A}}_q})}{\sqrt{\sum_{j=1}^{M}(A(q, e_j) - \mu_{\mathbf{A}q})^2 \sum_{j=1}^{M}(\bar{\alpha}(q, e_j) - \mu_{\bar{\mathbf{A}}_q})^2}} \tag{15}$$

Figure 3 visualizes $-\rho_q$; darker shades indicate a stronger tendency for higher intervention to correspond with lower accuracy (the desired adaptive behavior). The heatmap compellingly shows that for many question types, particularly when using robust uncertainty metrics like Ensemble and Entropy (which performed best in Table 2), there is a clear negative correlation. This provides strong evidence that our AUGI module intelligently directs greater debiasing effort towards areas where the model struggles most, validating the dynamic and targeted nature of our uncertainty-guided intervention.

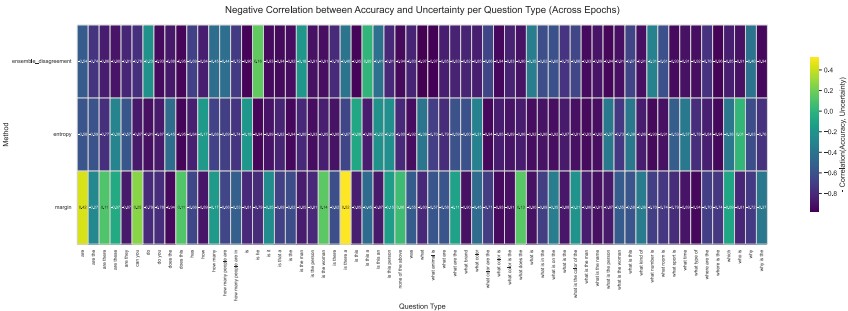

Figure 3: Heatmap visualizing the strength of negative correlation $(-\rho_q)$ between accuracy trajectory and average intervention strength trajectory for different question types and uncertainty methods. Higher values (e.g., darker shades) indicate that lower accuracy over time is more strongly associated with higher intervention.

### 4.5 ANALYSIS OF UNCERTAINTY-AWARE CURRICULUM LEARNING (UACL)

Figure 4 illustrates the impact of our Uncertainty-Aware Curriculum Learning strategy on training dynamics. We plot the validation performance (e.g., VQA-CP v2 accuracy) over training epochs for models trained: (i) without any curriculum learning, (ii) using UACL with an initial sample selection rate of 0.2 ($p_{init} = 0.2$), and (iii) using UACL with $p_{init} = 0.5$. For the UACL strategies, we compare learning curves where sample difficulty was evaluated using different uncertainty metrics (Entropy, Margin, Ensemble).

The results clearly demonstrate the benefits of UACL. Compared to training without a curriculum, UACL leads to more stable convergence and faster initial learning. Comparing $p_{init} = 0.2$ and $p_{init} = 0.5$, we observe that starting with a smaller, easier subset (0.2) results in smoother initial learning, while 0.5 converges slightly faster later on. The choice of uncertainty metric used for initial sorting shows minor differences in the curves, suggesting the core benefit comes from the uncertainty-aware scheduling itself rather than the specific sorting metric, provided it reasonably captures difficulty. Overall, these curves validate that UACL effectively leverages uncertainty to create a beneficial learning schedule, complementing the adaptive intervention mechanism.

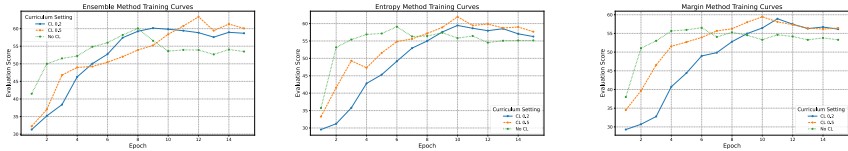

Figure 4: VQA-CP v2 validation accuracy during training. Comparing models trained without curriculum learning, with UACL ($p_{init} = 0.2$), and with UACL ($p_{init} = 0.5$), potentially showing curves for different uncertainty metrics used for sorting.

## 5 CONCLUSION

We introduced a novel VQA debiasing framework that synergistically integrates uncertainty estimation with causal counterfactual reasoning. Our architecture-agnostic approach features an Adaptive Uncertainty-Guided Intervention (AUGI) module that dynamically adjusts intervention strength based on model uncertainty, overcoming the limitations of static methods. This is complemented by an Uncertainty-Aware Curriculum Learning (UACL) strategy that aligns training with the model's adaptive capabilities. Extensive experiments and ablations on VQA-CP v2 and VQA v2 validate that our method significantly improves debiasing over state-of-the-art approaches while maintaining strong in-distribution performance.

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

## A   IMPLEMENTATION DETAILS

- **Backbones:** Our experiments demonstrate the architecture-agnostic nature of our proposed framework by integrating it with representative VQA backbones, including the Bottom-Up Top-Down (UpDn) architecture, as well as SAN and SMRL, as mentioned in the main paper. The core adaptive and causal mechanisms are built upon these foundational models.

- **Counterfactual Mechanism:** The counterfactual reasoning in our framework, which builds on established principles like those in CF-VQA, is implemented by intervening on the models internal representations. Rather than generating entirely new counterfactual input samples, our approach modifies internal pathways to compute the Natural Direct Effect of the Question ($NDE_Q$). For the baseline static counterfactual approach, this involves substituting certain vision-and-language and vision-only features with a non-informative constant. In our adaptive framework (AUGI), this intervention is dynamically adjusted by the uncertainty-derived adaptive factor $\alpha$, which interpolates between the original features and the non-informative constant, as described in Section 3.2 of the main paper.

- **Uncertainty Quantification (for AUGI module):** The Adaptive Uncertainty-Guided Intervention (AUGI) module employs several uncertainty estimation techniques, as detailed in Section 3.2 of the main paper. These include:

  - **Predictive Entropy** ($U_H$)**:** Calculated from the softmax probability distribution of the models output logits, normalized by the logarithm of the number of answer classes.
  - **Prediction Margin** ($U_{\text{Margin}}$)**:** Determined by the difference between the probabilities of the top two predicted answers.
  - **Ensemble Disagreement** ($U_{\text{Ens}}$)**:** Assessed by the variance in predictions from multiple independent prediction heads within the model. Each head typically consists of a small multi-layer perceptron (MLP) processing combined logit information. The variance across these headsoutputs for the adaptive factor $\alpha$ is then transformed into the final $\alpha$ value, often via another small MLP.

  The choice of internal model logits used for these uncertainty calculations can be configured to draw from various sources, such as the main VQA prediction branch, the question-only branch, or the vision-only branch.

- **Adaptive Factor** ($\alpha$)**:**

  - For entropy and margin-based uncertainty, the quantified uncertainty score (or scores, if multiple sources are used) is passed through a learnable transformation, typically a 2-layer MLP with a ReLU activation followed by a Sigmoid function, to produce the adaptive factor $\alpha \in [0, 1]$. The hidden dimensionality of this MLP is a configurable hyperparameter.
  - For ensemble-based uncertainty, $\alpha$ is derived from the variance of the ensemble membersoutputs, as detailed above.
  - To encourage meaningful adaptation and prevent $\alpha$ from collapsing to extreme values (0 or 1), a regularization term based on $\mathbb{E}[\alpha(1 - \alpha)]$ (Equation 8 in the main paper) is incorporated into the loss function, with a tunable weight.

- **UACL (Uncertainty-Aware Curriculum Learning):** The UACL strategy, as described in Section 3.3, is enabled through a specific configuration.

  - **Initial Difficulty Assessment:** The difficulty of each training sample is initially assessed based on a chosen metric. This can be an uncertainty-based metric, such as the predictive entropy calculated from an initial pass of the untrained or partially trained model over the dataset, or a bias-aware metric, such as one inversely related to pre-calculated answer biases for the samples' question type. Samples are then sorted by this difficulty score, typically from easier to harder.
  - **Curriculum Pacing:** A pacing function (e.g., linear) determines the fraction of the sorted training data to be used at each training epoch. This fraction typically increases from a specified start percentage in the initial epochs to include the full dataset by the final epochs, as outlined in Section 3.3.

- **Training:**

- **Optimizer:** Models are trained using the Adamax optimizer with its default learning rate.
- **Batch Size:** The batch size is set to 512.
- **Loss Function (for the adaptive model):** The total loss for our adaptive causal model is a weighted sum of several components, as detailed in Section 3.4 (Equation 13) of the main paper. This includes:
  * A primary cross-entropy loss on the models final (total effect) prediction.
  * Cross-entropy losses for the question-only and vision-only branches, with configurable weights.
  * The KL divergence loss $L_{KL}$ (Equation 12) to encourage consistency between the total effect and the (adaptively formulated) natural direct effect of the question.
  * The $\alpha$-regularization term $L_{\alpha\_reg}$ (Equation 8) to promote effective adaptation.

  Models are trained for 15 epochs, with evaluation performed on a validation set after each epoch. A fixed random seed of 1111 is used for reproducibility.

## B  ADDITIONAL RESULTS AND ANALYSIS

In this appendix, we provide additional experimental results and analyses that complement the main findings presented in the paper.

### B.1  QUALITATIVE ANALYSIS OF THE ADAPTIVE INTERVENTION MECHANISM

To provide a more intuitive understanding of our framework, we present a qualitative analysis of its behavior on examples with different types of bias, as illustrated in Figure 5. We first show a case where the foundational causal framework effectively handles language bias. We then use a "qualitative ablation"—manually setting adaptive factors ($\alpha_V$ or $\alpha_{VQ}$) to 0—to demonstrate how our adaptive intervention is crucial for overcoming more complex visual and multimodal biases where static approaches fail. This analysis provides a theoretical grounding for our observations by connecting them to the core equations presented in Section 3.

**Case 1: Handling Language Bias.** The top row of Figure 5 shows an example with a strong language prior ("banana" $\rightarrow$ "yellow"). The ground truth is "green". Our model correctly answers "green" even with its adaptive pathways ablated (i.e., when $\alpha_V = 0$ and $\alpha_{VQ} = 0$). This is because in this ablated state, the debiased logit defaults to the static baseline intervention:

$$L'_{debiased} = z_{TE} - F(c, Z_Q, c) \tag{16}$$

This result is consistent with prior work (e.g., CF-VQA) and demonstrates that for certain straightforward language biases, the baseline counterfactual mechanism provides a sufficient and robust foundation for debiasing.

**Case 2: Overcoming Vision Bias.** The middle row presents a more challenging case of visual bias. The question is "Do you see a beach?", where the image contains a wide water interface that creates a strong visual shortcut to "yes". The full AUGI model, using its adaptive debiasing logit, correctly answers "no". However, when we manually set $\alpha_V = 0$, we disable the adaptive intervention on the vision pathway. The debiased logit becomes:

$$L'_{debiased} = z_{TE} - F(Z'_{VQ}, Z_Q, c) \tag{17}$$

By forcing the vision feature $Z_V$ to be replaced by a non-informative constant $c$, this static intervention fails to account for the strong visual bias encoded in $Z_{VQ}$, leading to an incorrect answer ("yes"). In contrast, the full AUGI model learns a high $\alpha_V$, ensuring that the veridical visual evidence from $Z_V$ is preserved in the counterfactual calculation ($Z'_V \approx Z_V$), which is essential to override the bias.

**Case 3: Overcoming Multimodal Bias.** The bottom row shows a case of multimodal bias where the combination "man" + "wave" + "surfboard" creates a shortcut to "surfing". The man is actually "holding" the board. The full AUGI model correctly identifies this nuanced action. When we set $\alpha_{VQ} = 0$, we disable adaptivity for the joint vision-question pathway. The debiased logit is then calculated as:

$$L'_{debiased} = z_{TE} - F(c, Z_Q, Z'_V) \tag{18}$$

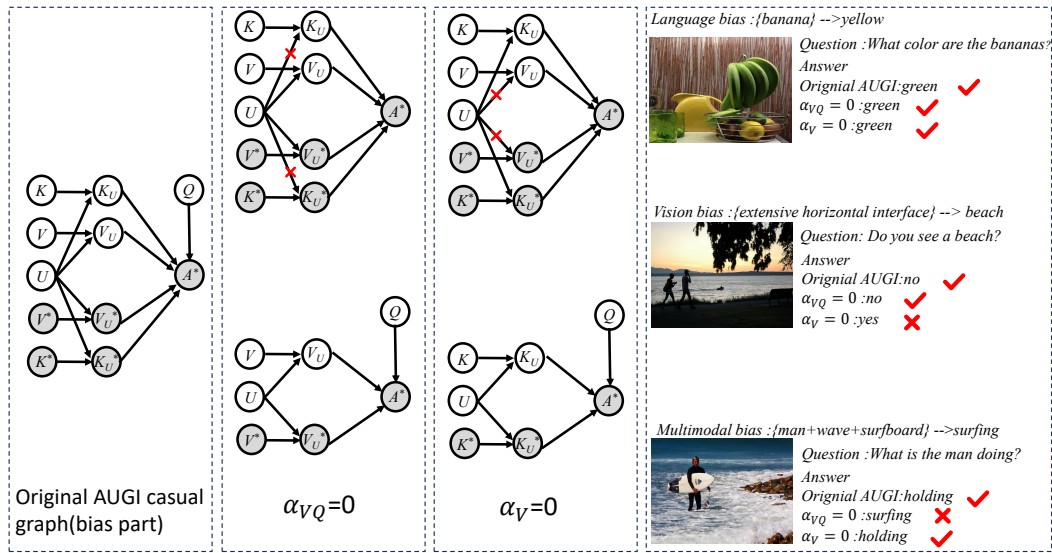

Figure 5: Qualitative analysis of the Adaptive Uncertainty-Guided Intervention (AUGI) mechanism. We demonstrate how our framework handles different types of bias and conduct a qualitative ablation study by manually setting pathway-specific adaptive factors to zero. This reveals where baseline causal methods suffice and where our adaptive interventions are critical. (**Top**) A *language bias* example, where the underlying causal framework is sufficient to overcome the "banana → yellow" shortcut. (**Middle**) A *vision bias* example, where the full model succeeds in overcoming the visual shortcut associating a wide water interface with a "beach", but disabling vision adaptivity ($\alpha_V = 0$) causes it to fail. (**Bottom**) A *multimodal bias* example, where the full model correctly identifies the nuanced action "holding", but disabling multimodal adaptivity ($\alpha_{VQ} = 0$) causes it to fail, proving the necessity of the adaptive component for complex shortcuts.

This intervention, by replacing the rich, contextual multimodal feature $Z_{VQ}$ with the constant $c$, discards the crucial information that the man is only \*holding\* the surfboard. The model then succumbs to the simplistic bias and predicts "surfing". The full AUGI model, by learning an appropriate $\alpha_{VQ}$, preserves this essential joint information in the counterfactual term, allowing it to correctly disambiguate the scene.

Together, these cases illustrate that while baseline causal methods can handle simple language priors, our adaptive, pathway-specific interventions are critical for robustly overcoming the more challenging visual and multimodal biases prevalent in VQA.

## B.2 Qualitative Case Studies

Traditional VQA models, while sometimes achieving high scores on benchmark datasets, often do so by exploiting strong language priors rather than engaging in robust multimodal reasoning. This means they can appear to perform well when answers align with these linguistic shortcuts (effectively handling some forms of language bias by memorization) but falter significantly when faced with visual biases (where the image contradicts the prior) or scenarios requiring genuine compositional understanding of both visual and textual modalities. This reliance on superficial correlations, as discussed in the main paper (Sections 1 and 2.1), is a key challenge our work addresses.

This section presents qualitative case studies to visually demonstrate how our proposed Adaptive Uncertainty-Guided Intervention (AUGI) framework, particularly when combined with Uncertainty-Aware Curriculum Learning (UACL), overcomes these limitations. Figure 6 showcases several examples from the VQA-CP v2 dataset, comparing the predictions of our full model (referred to as "Ours") against a standard VQA backbone (e.g., UpDn, labeled "Baseline") and a representative static causal debiasing method like CF-VQA. In these visualizations, it is crucial to note that red bars or

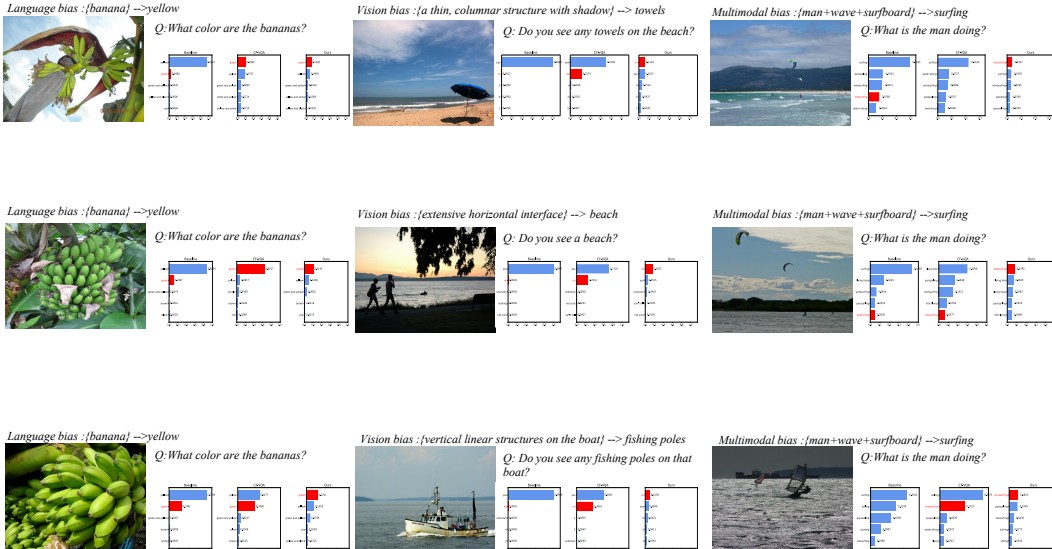

Figure 6: Qualitative examples from VQA-CP v2 comparing our method ("Ours") against a Baseline VQA model and a static counterfactual method (CF-VQA). Red bars/elements in the visualizations indicate the ground truth answers. Predicted answer distributions or top answers are shown for each model, illustrating differences in visual grounding and bias robustness.

highlighted elements consistently represent the ground truth answer, providing a clear reference for correctness.

The examples are chosen to highlight scenarios where common linguistic biases can lead non-adaptive models astray. For instance, consider cases where a question might trigger a strong but incorrect linguistic prior (e.g., asking about the color of an atypically colored object, like a non-yellow banana, if such an example is present in the figure). A traditional baseline model, heavily reliant on the learned Q-A correlation, might default to the high-frequency prior, thus failing to align with the red ground truth indicator. Static counterfactual methods like CF-VQA might offer partial improvement by attempting to negate the direct linguistic influence, but their fixed intervention strategy can be insufficient if the bias is particularly strong or visual cues are ambiguous, still leading to errors.

In contrast, our AUGI module, by dynamically assessing instance-specific uncertainty, modulates the counterfactual intervention more effectively. As seen in Figure 6, "Ours" more frequently aligns its prediction with the red ground truth. When the model detects high uncertainty (e.g., a conflict between visual evidence and a strong prior), it can apply a stronger, more targeted intervention. If it is confident in its visually grounded assessment, the intervention is more subtle. This adaptability is key to correctly identifying attributes even when they deviate from strong priors, leading to answers that are better grounded in the visual evidence.

## B.3 HYPERPARAMETER SENSITIVITY ANALYSIS

Our framework builds upon the causal baseline, inheriting hyperparameters like $\lambda_{KL}$ and $\lambda_{path}$, which we keep consistent with the original work for fair comparison. We introduce several new hyperparameters: the regularization weight $\lambda_{reg}$ (Eq. 8), the curriculum pacing exponent $p$ (Eq. 11), and the learnable parameters $\gamma$ and $\delta$ of the uncertainty mapping function (Eq. 4). To ensure rigor and reproducibility, this section details their management and analyzes the model's sensitivity to their values.

**Management of $\gamma$ and $\delta$:** The parameters $\gamma$ (steepness) and $\delta$ (center point) in the scaled sigmoid function are crucial for mapping the normalized uncertainty score $U_{norm}$ to the adaptive factor $\alpha$. To guarantee that higher uncertainty corresponds to a stronger intervention (i.e., a monotonically increasing relationship), we do not learn $\gamma$ directly. Instead, we learn an unconstrained underlying

parameter $\tilde{\gamma}$ and set $\gamma = \text{softplus}(\tilde{\gamma}) > 0$. The resulting $\gamma$ and $\delta$ are updated via backpropagation along with other network weights. We initialize $\tilde{\gamma}$ such that $\gamma \approx 1.0$ and set $\delta = 0.0$, which yields a standard sigmoid mapping at the start of training and is then fine-tuned by the model to best suit the specific data distribution and uncertainty characteristics.

**Role of $\lambda_{reg}$:** The regularization term $L_{\alpha\_reg}$ prevents $\alpha$ from collapsing to the trivial states of 0 or 1, encouraging the model to make genuine, nuanced adaptations. A very small $\lambda_{reg}$ might not provide a strong enough signal to prevent collapse, while a very large value could overly penalize confident predictions (where $\alpha$ should be close to 0), hindering learning.

**Sensitivity Analysis:** We conducted a sensitivity analysis on the VQA-CP v2 dataset using the UpDn backbone with Ensemble-based uncertainty to evaluate the impact of our key new hyperparameters: the regularization weight $\lambda_{reg}$, the curriculum pacing exponent $p$, and the number of heads $M$ for the ensemble uncertainty metric. The results, summarized in Table 3, show that our model's performance is stable across a reasonable range of values. For instance, performance remains high for $\lambda_{reg}$ between 0.01 and 0.1, and for pacing exponent $p$ between 1.0 (linear pacing) and 3.0. The number of ensemble heads $M$ shows optimal performance at $M = 5$, with diminishing returns or slight degradation for higher values, confirming it as a suitable trade-off between performance and complexity. This demonstrates the robustness of our approach. Our final model uses $\lambda_{reg} = 0.05$, $p = 2.0$, and $M = 5$, which were selected based on validation performance during this analysis.

Table 3: Sensitivity analysis for key hyperparameters on VQA-CPv2 using the UpDn backbone with Ensemble uncertainty. The default values used in our final model are marked with an asterisk (*). Performance is reported as overall accuracy (%).

| Sensitivity to $\lambda_{reg}$ | | Sensitivity to Exp. $p$ | | Sensitivity to Heads $M$ | |
| --- | --- | --- | --- | --- | --- |
| **Value** | **Acc. (%)** | **Value** | **Acc. (%)** | **Value** | **Acc. (%)** |
| 0.005 | 62.89 | 0.5 | 62.55 | 3 | 62.91 |
| 0.01 | 63.15 | 1.0 (Linear) | 63.02 | 5* | **63.36** |
| 0.05* | **63.36** | 2.0* | **63.36** | 8 | 63.28 |
| 0.1 | 63.21 | 3.0 | 63.18 | 10 | 63.25 |
| 0.5 | 61.98 | 4.0 | 62.75 | - | - |

### B.4 PERFORMANCE ON TRANSFORMER-BASED BACKBONES

To demonstrate the architecture-agnostic nature of our framework, we also conducted experiments integrating our method with large pre-trained transformer-based backbones, specifically BLIPLi et al. (2022; 2023) and LXMERTTan & Bansal (2019). For experiments involving BLIP, we follow the methodology of PW-VQA for fair comparison. For both BLIP-1 and BLIP-2 models, we use their pre-trained modules to extract image and text features. An attention layer is then used to obtain multimodal features, which are concatenated with the separate question and image features from the BLIP encoders. This final joint feature representation is fed to the classification layer. The results are presented in Table 4.

As shown in the table, our method consistently improves performance over the baseline backbones and other methods. When integrated with LXMERT, our approach achieves 75.16% on VQA-CPv2, significantly outperforming the base LXMERT model (44.56%) and achieving competitive results against other state-of-the-art methods like TPCL (77.23%). Similarly, with BLIP backbones, our method demonstrates substantial gains over the baselines. These results confirm that our uncertainty-guided adaptive intervention is not limited to traditional VQA architectures but is also effective when applied to powerful transformer-based models.

The control data in this table for other methods (e.g., PW-VQA, D-VQA, LSP) is drawn from either our own reproduction of the corresponding research or the data provided in the original papers. An asterisk (*) in the table indicates that the corresponding data was not provided in the original publication, and the code was not open-sourced, making it difficult for our team to reproduce the result.

Table 4: Performance comparison on VQA-CPv2 and VQAv2 using Transformer-based backbones (BLIP, LXMERT). Scores are reported in percentages (%).

| Model | Backbone | VQA-CPv2 (%) | | | | VQAv2 (%) | | | |
|---|---|---|---|---|---|---|---|---|---|
| | | All | Y/N | Num | Other | All | Y/N | Num | Other |
| *Backbone: BLIP* | | | | | | | | | |
| BLIP-1 | - | 34.42 | 54.97 | 14.48 | 35.10 | 54.59 | 70.70 | 40.00 | 46.21 |
| BLIP-2 | - | 34.93 | 52.21 | 15.15 | 36.10 | 61.25 | 79.12 | 48.77 | 49.00 |
| PW-VQA | BLIP-1 | 49.53 | 84.36 | 45.38 | 33.24 | 45.56 | 61.48 | 27.39 | 38.42 |
| PW-VQA | BLIP-2 | 45.84 | 85.17 | 19.16 | 32.73 | * | * | * | * |
| OURS | BLIP-1 | 52.66 | 83.77 | 47.58 | 34.76 | 50.21 | 66.92 | 30.12 | 41.10 |
| OURS | BLIP-2 | 50.27 | 86.59 | 40.66 | 35.00 | 59.31 | 78.91 | 42.00 | 46.25 |
| *Backbone: LXMERT* | | | | | | | | | |
| LXMERT | - | 44.56 | 57.62 | 23.34 | 49.88 | 51.77 | 55.69 | 21.66 | 60.12 |
| D-VQA | LXMERT | 69.75 | 80.43 | 58.57 | 69.75 | 64.96 | 82.18 | 44.05 | 57.54 |
| LSP | LXMERT | 71.06 | 86.56 | 59.01 | 66.24 | * | * | * | * |
| HCCL | LXMERT | 69.69 | 80.36 | 59.17 | 66.99 | * | * | * | * |
| TPCL | LXMERT | 77.23 | 93.10 | 72.00 | 70.34 | 78.03 | 93.34 | 65.11 | 69.81 |
| OURS | LXMERT | 75.16 | 91.23 | 69.73 | 72.01 | 73.28 | 89.61 | 62.33 | 70.00 |

## B.5 COMPUTATIONAL RESOURCE CONSUMPTION ANALYSIS

To discuss the practical trade-offs of our proposed methods, we conducted an analysis of their computational overhead. We trained the model 10 times and took the average to get the total training time; each time, the epoch was set to 15. For the Ensemble method, the number of lightweight prediction heads was set to five. Inference time was benchmarked by averaging over 10,000 validation samples. All experiments were based on the UpDn backbone on VQA-CPv2 with UACL enabled.

The analysis in Table 5 demonstrates that all proposed uncertainty metrics are computationally efficient. The performance gains from the Ensemble method, which achieves the highest accuracy, come at a very modest additional cost in training time, inference time, and GPU memory. This makes it a practical and effective choice for real-world applications.

Table 5: Computational Cost vs. Performance Analysis for Different Uncertainty Metrics. The time and memory usage results for margin and entropy closely approximate the tabulated data, though some discrepancies are observed. We attribute these variations to the experimental environment, as indicated by the $\sim$ symbol in the table.

| Uncertainty Method | Accuracy (%) | Total Training Time | Inference Time (per sample) | Added GPU Memory |
|---|---|---|---|---|
| Margin | 59.44 | 2 hrs (1.0x) | 20 ms (1.0x) | $\sim$10 MB |
| Entropy | 61.94 | 2 hrs (1.0x) | 20 ms (1.0x) | $\sim$10 MB |
| Ensemble ($M = 5$) | **63.36** | $\sim$2.1 hrs ($\sim$1.06x) | $\sim$22 ms ($\sim$1.1x) | $\sim$25 MB |

## B.6 EXTENDING THE FRAMEWORK TO LARGE VISION-LANGUAGE MODELS

To further test the scalability and generality of our framework, we extend it to modern large vision-language models (VLMs). In particular, we consider two representative 7B-parameter generative VLMs: LLaVA-v1.5 Liu et al. (2023a) and Qwen-VL-Chat (7B) Bai et al. (2023). These models differ from our standard UpDn-style backbones in that they are autoregressive generative decoders rather than classification heads.

Given this generative nature, we adapt the AUGI mechanism from a feature-level interpolation to an *adaptive loss* formulation. Instead of interpolating internal representations with a constant counterfactual vector, we introduce a counterfactual debiasing regularizer, $L_{debias\_reg}$, that penalizes reliance on the question-only pathway. A scalar uncertainty score is mapped to an adaptive factor $\alpha \in [0, 1]$, and this factor dynamically gates the strength of $L_{debias\_reg}$ during fine-tuning. Intuitively, higher uncertainty leads to a larger effective debiasing weight, preserving the core principle of our framework: when the model is less certain, it should rely less on shortcut pathways and more on robust visual grounding.

All VLM experiments are conducted with LoRA-based fine-tuning (rank $r = 16$) using $2 \times$ A6000 (48GB) GPUs. We report both OOD performance on VQA-CP v2 (test split) and ID performance on VQA v2 (validation split), together with training time, training VRAM usage, and inference latency. The results are summarized in Table 6.

Table 6: Performance and resource consumption when extending our framework to large vision-language models (VLMs). "SFT" denotes standard supervised fine-tuning. OOD performance is measured on VQA-CP v2 (test), ID performance on VQA v2 (val). All fine-tuning uses LoRA with rank $r = 16$ on $2 \times$ A6000 (48GB) GPUs. Inference latency is measured on a single A6000 GPU. Bold indicates the best OOD (VQA-CP v2 overall) performance for each backbone.

| Model | Method | VQA-CP v2 Overall (OOD, %) | VQA-CP v2 Y/N (OOD, %) | VQA-CP v2 Num (OOD, %) | VQA-CP v2 Other (OOD, %) | VQA v2 Overall (ID, %) | Train Time (hrs) | Train VRAM (GB) | Infer Latency (ms) |
|---|---|---|---|---|---|---|---|---|---|
| *LLaVA-v1.5-7B* | | | | | | | | | |
| LLaVA-v1.5-7B | Baseline (Zero-Shot) | 31.52 | 40.11 | 12.89 | 34.05 | 70.33 | – | – | ~170 |
| | Baseline (SFT) | 40.21 | 51.20 | 14.55 | 43.10 | 65.12 | ~8.5 | ~30 | ~170 |
| | UACL-VLM (Ours) | 45.33 | 55.02 | 16.80 | 48.15 | 67.22 | ~8.0 | ~30 | ~170 |
| | AUGI-VLM (Ours) | 52.89 | 63.40 | 20.15 | 55.04 | 68.05 | ~16.0 | ~32 | ~170 |
| | **UACL + AUGI (Ours)** | **55.10** | **65.22** | **21.03** | **57.34** | **68.31** | **~15.5** | **~32** | **~170** |
| *Qwen-VL-Chat (7B)* | | | | | | | | | |
| Qwen-VL-Chat (7B) | Baseline (Zero-Shot) | 35.80 | 42.50 | 14.10 | 38.66 | 75.14 | – | – | ~185 |
| | Baseline (SFT) | 45.15 | 54.02 | 16.20 | 48.71 | 70.02 | ~9.5 | ~32 | ~185 |
| | UACL-VLM (Ours) | 50.82 | 58.11 | 18.90 | 54.40 | 72.15 | ~9.0 | ~32 | ~185 |
| | AUGI-VLM (Ours) | 58.44 | 66.20 | 23.51 | 61.30 | 73.01 | ~17.5 | ~34 | ~185 |
| | **UACL + AUGI (Ours)** | **61.05** | **68.33** | **25.14** | **63.88** | **73.25** | **~17.0** | **~34** | **~185** |

These results highlight several key observations. First, the relatively low zero-shot OOD accuracies (31.52% for LLaVA and 35.80% for Qwen-VL) confirm that even strong VLMs remain highly vulnerable to language priors on VQA-CP v2. Second, standard fine-tuning on the biased training set yields only modest OOD gains while substantially degrading ID performance on VQA v2, reflecting a form of catastrophic forgetting. By contrast, our full UACL + AUGI configuration delivers large OOD improvements over SFT (+14.9 and +16.0 absolute points for LLaVA and Qwen-VL respectively) while largely preserving ID performance. For Qwen-VL, the final ID accuracy (73.25%) even slightly exceeds the SFT baseline (70.02%) and approaches the original zero-shot score (75.14%). Finally, although the adaptive debiasing loss increases training time due to additional counterfactual passes, the LoRA-finetuned models do not incur extra inference latency compared to SFT, indicating that our framework is practical for real-world VLM deployment.

## B.7 DIAGNOSTIC EVALUATION ON THE VQA-VS DATASET

To further validate our framework and provide a more fine-grained analysis of its debiasing capabilities, we conduct additional experiments on the VQA-VSSi et al. (2022b) dataset. While VQA-CPv2 is an effective OOD benchmark for evaluating robustness against various biases, its distribution shift is primarily dominated by language priors(Although this dataset also contains multimodal bias and vision bias, these biases constitute a relatively small proportion and lack explicit definition—they were discovered through our experimental investigation). The VQA-VS dataset offers a more diagnostic evaluation by defining nine specific shortcut types spanning language-based (e.g., Question Type, Keyword), visual-based (e.g., Key Object, Key Object Pair), and multimodal domains (e.g., combinations of language and visual elements). This allows for a more surgical analysis of a model's robustness. For this dataset, we report performance on the standard In-Distribution (ID) test set and, more importantly, the mean accuracy across all nine Out-of-Distribution (OOD) shortcut-specific test sets, which we denote as OOD-mean. A high OOD-mean score indicates that a model is robust not just to a single type of bias, but to a diverse array of spurious correlations.

The results, presented in Table 7, show that our approach significantly outperforms existing methods. On both the UpDn and LXMERT backbones, our full framework (AUGI+UACL) achieves the highest performance on the critical OOD-mean metric, with scores of **60.75%** and **65.22%** respectively. This represents a substantial improvement over the vanilla backbones and even strong causal baselines like CFVQA. Notably, while some methods see a large drop in performance from ID to OOD, our method maintains a much smaller gap, demonstrating its effectiveness in mitigating shortcut learning and improving out-of-distribution generalization. These results confirm that our adaptive, uncertainty-guided interventions are effective not just against general language priors, but also against a wide array of specific visual and multimodal shortcuts.

Table 7: Performance comparison on the VQA-VS dataset. Our method demonstrates significant improvements over all baselines, particularly on the challenging OOD-mean metric, indicating superior debiasing and generalization capabilities across various shortcut types.

| Method | Backbone | ID (%) | OOD-mean (%) |
|---|---|---|---|
| updn | updn | 65.20 | 46.80 |
| LMH | updn | 56.89 | 45.85 |
| CF-VQA | updn | 59.12 | 49.33 |
| LPF | updn | 54.72 | 43.31 |
| OURS (AUGI) | updn | 62.59 | 57.89 |
| OURS (AUGI+UACL) | updn | 63.21 | **60.75** |
| LXMERT | LXMERT | 72.24 | 53.92 |
| LMH | LXMERT | 70.22 | 54.41 |
| CF-VQA | LXMERT | 67.22 | 56.30 |
| LPF | LXMERT | 68.48 | 50.83 |
| OURS (AUGI) | LXMERT | 71.35 | 64.20 |
| OURS (AUGI+UACL) | LXMERT | 71.50 | **65.22** |

## C    LIMITATIONS AND FUTURE DIRECTIONS

This paper pioneers dynamically adjusted causal interventions, a significant step beyond traditional methods that often rely on simplistic feature subtractions. However, a limitation is that our current dynamic fusion of counterfactual information, while adaptive via uncertainty, still operates within a framework of explicit mathematical operations. Furthermore, our framework introduces new hyperparameters related to the adaptive mechanism ($\lambda_{reg}$, $\gamma$, $\delta$) and curriculum strategy ($p$). While we demonstrate robustness to these in our sensitivity analysis (Appendix B.3), their optimal selection adds a layer of complexity compared to static methods.

Another limitation and avenue for future work relates to the static nature of our Uncertainty-Aware Curriculum Learning (UACL). The curriculum is fixed after an initial warm-up phase—a deliberate design choice for stability, as our preliminary explorations suggested that a dynamic curriculum that re-evaluates difficulty can be unstable. We hypothesize this is because the uncertainty signal evolves: early in training, it can be a noisy proxy for true, bias-related difficulty, while late in training, the model may become "confidently wrong" on biased samples, mislabeling them as easy. A static curriculum provides a stable, regularizing training signal, but this highlights the warm-up duration as a critical hyperparameter. Exploring this stability-adaptivity trade-off and developing hybrid static-dynamic curricula is a promising direction.

The intricate nature of multimodal debiasing, particularly in VQA, often mirrors challenges in understanding algorithmic "hallucinations" stemming from complex interactions between data distributions and modalities—a domain where fundamental principles are still evolving. Therefore, a crucial future direction involves exploring more heuristic and data-driven counterfactual generation schemes. For instance, recent explorations in Large Language Models, such as representing counterfactual branches by substituting different attention mechanisms Zhou et al. (2025), exemplify promising paths towards more intuitively grounded and flexible interventions that could further enhance robustness against subtle biases.

Finally, our uncertainty-based techniques have strong conceptual ties to other areas of machine learning. The uncertainty scores used in our AUGI module could be adapted for data-pruning strategies, helping to identify and remove noisy or uninformative samples from the training set. Similarly, our UACL strategy, which sequences data based on uncertainty, is closely related to coreset selection, where the goal is to find a small, representative subset of the data for efficient training. Exploring these connections and formally adapting our methods for data-efficient learning presents a valuable direction for future research.

