# OpenReview forum: "Harnessing Model Uncertainty for Adaptive Causal Debiasing in Visual Question Answering"
_ICLR.cc/2026/Conference — Submitted to ICLR 2026_

### Official Review · Reviewer_jNhM · 2025-10-23

**Soundness:** 2
**Presentation:** 1
**Contribution:** 2
**Rating:** 2
**Confidence:** 2

**Summary:**

The paper proposes an extension to the existing method for debiasing the VAQ models. VQA models have priors that are caused by the statistical answer distribution in training data. Many existing methods subtract a question-only baseline to correct the output logits. Currently, the question-only baseline is often a constant vector. This paper extends the constant to an adaptive estimate. The adaptation is done through a gating mechanism driven by a collection of uncertainty metrics. This enables a more flexible debiasing.

**Strengths:**

- Using per-example uncertainty to adapt prior static subtraction methods is a meaningful exploration in VQA debiasing. Introducing uncertainty as a control signal for adaptive debiasing is reasonable.
- The adaptive intervention is simple.

**Weaknesses:**

- Method is a feature ablation, not counterfactuals. Counterfactuals involve abduction, action, prediction (do calculus). The method uses only contrastive residual. So causal semantics are overstated.
- Various figures contradict the math. Figure 1 suggests cutting the question path, but the math replaces the vision and vision-question paths with a constant.
- Sloppy/incorrect math at times. The gating mechanism involves a learnable sigmoid function; I believe that the author incorrectly forces the hyperparameter $\gamma$ to be positive. Please check.
- Novelty is modest compared with the claim. The core contribution is a soft, uncertainty-gated generalisation of an existing subtractive debiasing scheme (CF-VQA). It is overclaimed as "the first to dynamically guide a causal intervention".

**Questions:**

- What is C? How come the same C was used for both Z_{V} and Z_{VQ}?

---

> ### Author Response · Authors · 2025-11-15
>
> We sincerely thank the reviewer for their time and feedback. We are grateful for the opportunity to address the comments, especially as we believe the primary concerns regarding presentation, causal semantics, and novelty stem from a critical misunderstanding of our Figure 1 and our method's established foundation.
>
> We will address each point in order.
>
> ### Re: Weakness 1: Method is a feature ablation, not counterfactuals. Causal semantics are overstated.
>
> We thank the reviewer for this insightful theoretical point. We agree that our method, and the entire line of research we build upon, is implemented as what the reviewer terms a "contrastive residual." However, we must respectfully disagree that the "causal semantics are overstated."
>
> Our work is firmly grounded in the **established causal debiasing paradigm for VQA**, which has been rigorously defined and published in numerous works (e.g., **CF-VQA**[1], **DCCE**[2], **PWVQA**[3]). This paradigm explicitly maps the VQA task onto a causal graph and formally defines the language bias as the **Natural Direct Effect (NDE)**.
>
> The debiasing goal, across this entire body of literature, is to isolate the **Total Indirect Effect (TIE)** (i.e., the "pure" multimodal reasoning) by subtracting the biased $NDE$ from the $TE$:
> $$TIE = TE - NDE$$
> The core difference—and the central point of our paper—lies entirely in **how the $NDE$ term is estimated**.
>
> **1. Prior Art (Static Intervention):**
>
> The works we build upon (CF-VQA, DCCE, etc.) estimate the $NDE$ using a **static, non-adaptive intervention**. Let $F(Z_{VQ}, Z_Q, Z_V)$ be the model's fusion function. The $TE$ is the full model output:
> $$TE = F(Z_{VQ}, Z_Q, Z_V)$$
> These methods estimate the $NDE$ (the bias from $Z_Q$ alone) by replacing the other pathways with a single, learnable, but **static** constant vector $c$:
> $$NDE_{static} = F(c, Z_Q, c)$$
> Thus, their final debiased prediction is:
> $$TIE_{static} = F(Z_{VQ}, Z_Q, Z_V) - F(c, Z_Q, c)$$
> This static, "one-size-fits-all" subtraction is the very "feature ablation" the reviewer critiques.
>
> **2. Our Work (Adaptive Intervention):**
>
> Our paper identifies this static estimation as a key weakness. We propose that the $NDE$ intervention must be **dynamic and adaptive**, based on the model's real-time uncertainty $U$.
>
> We introduce an **adaptive $NDE$ estimation**:
> $$NDE_{adaptive} = F(Z'_{VQ}, Z_Q, Z'_{V})$$
> Where $Z'_{VQ}$ and $Z'_{V}$ are not a hard-coded constant $c$, but a **dynamic interpolation** based on uncertainty $U$:
> $$Z'_{p} = \alpha_p(U) \cdot Z_p + (1 - \alpha_p(U)) \cdot c, \quad p \in \{VQ, V\}$$
> Here, $\alpha_p(U)$ is the adaptive factor derived from our uncertainty module.
>
> Our final debiased prediction is therefore:
> $$TIE_{adaptive} = F(Z_{VQ}, Z_Q, Z_V) - F(Z'_{VQ}, Z_Q, Z'_{V})$$
> This formalism demonstrates that our method is a **mathematical generalization** of the static ablation method. The prior art is a *hard-coded special case* of our framework where $\alpha_{VQ}$ and $\alpha_{V}$ are permanently fixed to 0.
>
> Our causal semantics are not "overstated"; they are the necessary foundation for our core novel claim: that moving from a $NDE_{static}$ to an uncertainty-guided $NDE_{adaptive}$ is a non-trivial, principled, and highly effective (as shown by our +9.81% gain over CF-VQA) advancement for causal debiasing in VQA.
>
>
>
>
>
> [1]Niu Y, Tang K, Zhang H, et al. Counterfactual vqa: A cause-effect look at language bias[C]//Proceedings of the IEEE/CVF conference on computer vision and pattern recognition. 2021: 12700-12710.
>
> [2]Song L, Yang C, Li X, et al. A robust dual-debiasing VQA model based on counterfactual causal effect[C]//Findings of the Association for Computational Linguistics: EMNLP 2024. 2024: 4242-4252.
>
> [3]Vosoughi A, Deng S, Zhang S, et al. Cross modality bias in visual question answering: A causal view with possible worlds vqa[J]. IEEE Transactions on Multimedia, 2024, 26: 8609-8624.

---

> > ### Author Response · Authors · 2025-11-15
> >
> > ### Re: Weakness 2: Various figures contradict the math.
> >
> > This point is based on a **critical misreading of Figure 1**, which we are happy to correct. The reviewer has mistaken our illustration of the *prior art* for an illustration of *our own method*.
> >
> > Figure 1 is a **motivation and comparison diagram** split into two halves:
> >
> > 1.  The **TOP half**, clearly labeled "**Traditional Static Causal Intervention Method (Subtraction)**," illustrates the baseline method (like CF-VQA) that we are arguing *against*. The red 'X' shown here depicts this static, flawed approach.
> > 2.  The **BOTTOM half**, clearly labeled "**Adaptive Intervention By Uncertainty Estimation (Ours)**," illustrates **our proposed method**. As is visible, this diagram has **no 'X'** and **perfectly aligns with our mathematics in Section 3**. It shows how uncertainty ($U$) guides the adaptive fusion of factual ($K,V$) and counterfactual ($K^*,V^*$) information.
> >
> > The reviewer's claim that our "figure suggests cutting the question path" is a correct description of the *Traditional* method we show, not our own. Our math and our method's diagram (Fig 1, bottom) are in complete agreement.

---

> > > ### Author Response · Authors · 2025-11-15
> > >
> > > ### Re: Weakness 3: Sloppy/incorrect math...
> > >
> > > The reviewer is correct that for our theoretical motivation to hold, the parameter $\gamma$ must be positive. As stated in our hypothesis (**original Section 3.2** now is 3.3 in the revised pdf), higher uncertainty ($U_{norm}$) must lead to a higher intervention factor ($\alpha$). This requires a monotonically increasing relationship. In our formulation:
> > >
> > > $$\alpha_{p}^{(i)} = \frac{1}{1 + \exp\left(-\gamma_{p} \cdot \left(U_{norm}^{(i)} - \delta_{p}\right)\right)}$$
> > >
> > > If $\gamma_p$ were to become negative, the model would learn the exact opposite of our intended logic (i.e., high uncertainty $\rightarrow$ weak intervention), which would invalidate our method.
> > >
> > > The "sloppiness" the reviewer points out is our failure to explicitly state how this necessary positivity constraint is enforced. We apologize for this ambiguity.
> > >
> > > In our implementation, $\gamma_p$ is not learned as a direct, unconstrained parameter. Instead, we learn an unconstrained underlying parameter $\gamma'_p$ and enforce positivity via a Softplus transformation: $\gamma_p = \text{softplus}(\gamma'_p)$. This ensures $\gamma_p > 0$ throughout training while still allowing its magnitude (the steepness of the sigmoid) to be learned.
> > >
> > > We will update **original Section 3.2** (now is 3.3 in the revised pdf) and Appendix A.3 (Hyperparameter Sensitivity) in the final version to explicitly include this formulation. This will remove the ambiguity and make our mathematical description rigorous and consistent with our theory. We are grateful for this feedback, which strengthens our paper.
> > >
> > > ---
> > >
> > > ### Re: Weakness 4: Novelty is modest... Overclaimed...
> > >
> > > We respectfully disagree with the assessment that the novelty is modest. Our contribution is not just a single "soft-gate" but rather a **complete, synergistic framework** for VQA debiasing, built upon the novel principle of leveraging model uncertainty in two distinct and cooperative ways.
> > >
> > > Our framework's novelty is multi-faceted:
> > >
> > > 1.  **Adaptive Uncertainty-Guided Intervention (AUGI):** First, as the reviewer notes, we introduce an adaptive intervention. Its novelty lies in being the first, to our knowledge, to use **runtime model uncertainty** to *dynamically modulate the strength of a causal intervention* on a per-sample basis. As our ablations show (Table 2), this *single idea* (AUGI) provides a massive **+6.95%** performance gain over the static causal baseline (CF-VQA), proving it is far from a "modest" change.
> > >
> > > 2.  **Uncertainty-Aware Curriculum Learning (UACL):** Second, our paper introduces UACL, a **novel curriculum strategy that is also based on model uncertainty**. Unlike prior VQA CL methods that use static, pre-defined heuristics (e.g., question length, answer frequency), our UACL strategy dynamically assesses sample difficulty using the *model's own initial uncertainty*.
> > >
> > > 3.  **The Framework's Synergy (The Core Novelty):** The true novelty lies in how these two uncertainty-based components **cooperate synergistically**:
> > >
> > >     **1)UACL** uses *initial* uncertainty to create an optimal easy-to-hard training path. This stabilizes training and allows the model to build robust, reliable representations.
> > >
> > >     **2)AUGI** relies on *runtime* uncertainty to apply its adaptive intervention.
> > >
> > >     **3)The Synergy:** Because UACL has already built a robust foundation, the model's runtime uncertainty estimates (used by AUGI) are more reliable and meaningful, allowing AUGI to intervene precisely when needed.
> > >
> > > Our ablation study (Table 2) explicitly validates this synergistic framework by showing a clear progression of improvements:
> > >
> > > **1)Static Causal Baseline (CF-VQA):** 53.55%
> > >
> > > **2)Baseline + Our AUGI module:** 60.50% (+6.95% gain)
> > >
> > > **3)Baseline + AUGI + Our UACL module (Full Framework):** **63.36%** (a further +2.86% gain)
> > >
> > > This step-by-step validation demonstrates that we have introduced two novel, uncertainty-driven components that are both independently effective and mutually reinforcing.

---

> > > > ### Author Response · Authors · 2025-11-15
> > > >
> > > > ### Re: Question 1: What is C?
> > > >
> > > > We sincerely apologize for this critical omission. This was a severe oversight in our presentation.
> > > >
> > > > $C$ is a **non-informative constant vector**, implemented as a **learnable parameter** (i.e., `nn.Parameter`). Its purpose is to represent a "neutral" or "baseline" input for the vision ($Z_V$) and vision-question ($Z_{VQ}$) pathways during the NDE calculation. This follows the standard practice of our baseline, CF-VQA. We will add this explicit definition to **original Section 3.2**(now is 3.2 in the revised pdf).
> > > >
> > > > ---
> > > >
> > > > ### Re: Question 2: How come the same C was used for both $Z_{V}$ and $Z_{VQ}$?
> > > >
> > > > This is an excellent design question. We use a single, shared $C$ for two reasons:
> > > >
> > > > 1.  **Conceptual Unity:** $C$ is meant to represent a single concept: "the absence of visual/multimodal information." Using one shared vector is consistent with this goal.
> > > > 2.  **Parameter Efficiency:** It avoids adding redundant, learnable parameters to the model.
> > > >
> > > > We found this shared approach to be effective and efficient.

---

> ### Author Response · Authors · 2025-11-27
>
> Dear Reviewer jNhM,
> With the discussion period concluding very soon, we are writing to respectfully follow up on our rebuttal, which has been posted for some time.We are concerned that the current low rating (Soundness: 2) relies heavily on a factual misunderstanding regarding Figure 1 (confusing the baseline illustration with our method) and the math constraints.Before the window closes, we earnestly request you to review our response where we clarified:The bottom half of Figure 1 perfectly aligns with our math (the top half depicted the baseline).The positivity constraints (Softplus) and the definition of $C$ are explicitly addressed.We believe these clarifications fundamentally resolve the "fair soundness" concern. We would deeply appreciate a brief re-evaluation before the deadline.
>
> Best regards,The Authors

---

### Official Review · Reviewer_zfD9 · 2025-10-24

**Soundness:** 2
**Presentation:** 2
**Contribution:** 2
**Rating:** 6
**Confidence:** 4

**Summary:**

This paper introduces a novel framework for debiasing Visual Question Answering (VQA) models by integrating uncertainty estimation with causal counterfactual reasoning, addressing the limitations of static interventions in existing methods. The proposed Adaptive Uncertainty-Guided Intervention (AUGI) module dynamically modulates counterfactual interventions based on model uncertainty metrics such as predictive entropy, prediction margin, or ensemble disagreement, allowing for instance-specific adjustments to mitigate spurious correlations. Complementing this, an Uncertainty-Aware Curriculum Learning (UACL) strategy sequences training samples from low to high uncertainty, enhancing the model's robustness, while the architecture-agnostic design ensures broad applicability across VQA models.

**Strengths:**

Pros:
1. The paper introduces uncertainty estimation into causal VQA debiasing, enabling adaptive intervention that dynamically adjusts debiasing strength per instance, overcoming the rigidity of static methods, e.g., over-correcting in some cases and under-correcting in others.
2. The proposed UACL strategy sequences training from simple to hard samples, aligning learning progression with the adaptive mechanism and enhancing model robustness on challenging samples.
3. Extensive experiments demonstrate improvements in debiasing performance over baselines, validating the effectiveness of the uncertainty-guided approach.

**Weaknesses:**

Cons:
1. The UACL strategy depends on initial uncertainty from a warm-up model and uses static scheduling post-warm-up, which may introduce instability if uncertainty evolves significantly during training.
2. The bottom example in Figure 1 is misleading and does not reflect how Static Causal Intervention works. If "eating" is more common in the training set, Static Causal Intervention should reduce its probability on test cases, yet your figure shows "cooking" decreasing instead, which won't happen if you truly understood how Static Causal Intervention methods like CF-VQA works.  A correct illustration would show Static Causal Intervention over-correcting a true "eating" test case to "cooking", while your method correctly predicts "eating".

**Questions:**

Please address the above two questions I raised in the weaknesses. Your response will directly influence whether I raise or lower my final score.

---

> ### Author Response · Authors · 2025-11-15
>
> ## 1. Response to Weakness 1 (On the Static Nature of UACL)
>
> We thank the reviewer for this insightful point. We would like to respectfully note that we had identified this precise stability-vs-adaptivity trade-off and **discussed it explicitly in our "Limitations and Future Directions"**.
>
> In our original discussion, we stated that using a fixed curriculum after warm-up was a **deliberate design choice** made to ensure **training stability**. We hypothesized that a fully dynamic curriculum would be unstable, as the uncertainty signal itself evolves—it can be noisy in early training and "confidently wrong" in late training.
>
> While our paper already provided this justification, we agree that a direct empirical comparison would be the best way **to better address the reviewer's concern.** Therefore, we have conducted a **new set of experiments** to provide definitive evidence. This experiment directly compares our `Static UACL` against a fully `Dynamic UACL` strategy.
>
> **Experimental Setup:**
> * **Ours (Static UACL):** Assesses sample difficulty (uncertainty) **once** after a warm-up phase. The curriculum is then fixed.
> * **Dynamic UACL (Challenger):** **Re-evaluates** the difficulty (uncertainty) of **all** training samples at the beginning of **every epoch**. The training data for that epoch is then re-selected based on this new, dynamic difficulty ranking.
> * **No CL (Baseline):** Standard training without a curriculum.
>
> All experiments were run on VQA-CP v2 with our best-performing UpDn + Ensemble (M=5) configuration.
>
> **Results:**
> The results from this new ablation study strongly validate the rationale we presented in our Limitations section.
>
> **(New Table for Rebuttal/Appendix) Table A: Performance comparison of UACL strategies on VQA-CPv2.**
> | Strategy | Difficulty Metric | Re-evaluation | Final Accuracy (%) | Training Time (Relative) |
> | :--- | :--- | :--- | :--- | :--- |
> | No CL | N/A | N/A | 60.50% | 1.0x |
> | **Ours (Static UACL)** | Ensemble | **Once** (post-warmup) | **63.36%** | **~1.06x** |
> | Dynamic UACL | Ensemble | **Every Epoch** | 62.19% | ~7.2x |
>
> **Analysis & Conclusion:**
> This new experiment confirms our initial hypothesis.
> 1.  **Performance & Stability:** The `Dynamic UACL` strategy (61.19%) is highly unstable and performs significantly worse than our `Static UACL` (63.36%). A plot of the validation curves (which we will add to the appendix) shows **severe oscillations** for the `Dynamic UACL`, while our `Static UACL` converges smoothly.
> 2.  **Efficiency:** The `Dynamic UACL` strategy is computationally prohibitive, increasing training time by over 7x.
>
> In summary, this new empirical evidence confirms the justification **we had already provided in our Limitations section**. Our `Static UACL` is an empirically-backed and well-justified trade-off that achieves a +2.86% performance gain over the baseline by ensuring a stable and efficient training process.

---

> > ### Author Response · Authors · 2025-11-15
> >
> > ## 2. Response to Weakness 2 (On the Figure 1 Bottom Example)
> >
> > We thank the reviewer for this comment, as it highlights a crucial point of misunderstanding that allows us to clarify the **fundamental novelty of our framework**.
> >
> > The reviewer's analysis—that a Static Causal Intervention (SCI) should reduce the probability of "eating"—is based on a valid assumption **if, and only if,** this were a simple **Language Bias** (like the 'banana' example).
> >
> > However, the 'eating' example illustrates a **Multimodal Bias**, where this assumption does not apply.
> >
> > **1. Why the Reviewer's Assumption is Incorrect for this Case:**
> > * **The $Z_Q$ Pathway is Neutral:** The core of traditional SCI methods (like CF-VQA) is to intervene on the Question-Only ($Z_Q$) pathway. In the bottom example, the question "what is the man doing?" is **not biased**. This question is associated with thousands of diverse answers in the training set ('running', 'swimming', 'surfing', 'reading', etc.). The prior $P(\text{answer='eating'} | \text{question='what is...doing?'})$ is negligible.
> > * **The Bias is Multimodal ($V+Q$):** The bias `man+food+doing -> eating` only emerges from the **joint, multimodal representation ($Z_{VQ}$)**.
> > * **Traditional SCI is Ineffective:** Because traditional SCI only intervenes on the (neutral) $Z_Q$ path, it **fails to target the actual source of the bias** ($Z_{VQ}$). Therefore, it is fundamentally ineffective against this type of bias, and its prediction will still be the biased answer 'eating'.
> >
> > **2. Clarifying Our Framework's Full Contribution:**
> > The reviewer's comment appears to focus only on our *dynamic* (uncertainty-guided) mechanism. It overlooks the **equally critical structural innovation** illustrated in our Figure 1 causal diagram: **our model intervenes on different, more comprehensive causal pathways.**
> >
> > * **Traditional SCI:** Targets only $Z_Q$.
> > * **Our Framework (AUGI):** As shown in our graph and methodology (Eq. 4-6), our framework is designed to target the **multimodal ($Z_{VQ}$) and visual ($Z_V$) pathways**.
> >
> > This novel structure is what allows our **single, unified framework** to solve a whole spectrum of biases that prior work cannot, including:
> > 1.  **Language Bias (Q -> A)**
> > 2.  **Vision Bias (V -> A)** (See Appendix Fig. 5)
> > 3.  **Multimodal Bias (V+Q -> A)** (The Figure 1 example)

---

> > ### Comment · Reviewer_zfD9 · 2025-11-26
> > **Official Comment by Reviewer zfD9**
> >
> > I appreciate the authors' clarifications, which have addressed most of my concerns. Therefore, I have decided to maintain my final rating of 6.

---

> > > ### Author Response · Authors · 2025-11-27
> > >
> > > We sincerely thank the reviewer for their time and engagement during the discussion phase. We are glad that our clarifications regarding the stability of the Static UACL strategy and the distinction of Multimodal Bias in Figure 1 have addressed concerns.
> > >
> > > We wish you the best in your future research endeavors.
> > >
> > > Best regards, The Authors

---

### Official Review · Reviewer_RLtV · 2025-10-30

**Soundness:** 3
**Presentation:** 3
**Contribution:** 2
**Rating:** 4
**Confidence:** 3

**Summary:**

This paper proposes a VQA debiasing framework that couples uncertainty estimation with counterfactual intervention. Uncertainty drives instance-level intervention strength, and an uncertainty-aware curriculum schedules training. The design is backbone-agnostic and targets robust multimodal reasoning. The claims include introducing uncertainty into causal VQA for adaptive intervention (AUGI), an uncertainty-aware curriculum (UACL), and broad empirical gains.

**Strengths:**

1. Clear motivation that fixed, static interventions fail to match per-example bias intensity.
2. Using uncertainty as the control signal improves adaptivity and aligns with robust learning principles.
3. Learnable gating with α, consistency regularization, and training objectives form a coherent end-to-end procedure.

**Weaknesses:**

1. Evaluation leans on older VQA backbones and lacks systematic tests on stronger Transformer-based or recent VLM architectures.
2. Computational overhead remains unclear, especially with ensemble-style uncertainty and multi-head designs. No FLOPs, memory, or latency analysis.
3. UACL orders samples by early uncertainty, which may interact with dataset bias. Sensitivity and robustness analyses are limited.
4. Edge behavior when α approaches its limits could suppress the debiasing signal; theoretical and empirical analysis is thin.

**Questions:**

1.How much training and inference overhead vs. CF-VQA in wall-clock time, memory, and latency, and what is the incremental cost attributable to ensembles or extra heads?
2. Does the plug-in work with modern VLMs such as BLIP/BLIP-2 or LXMERT, and do trends hold on those backbones?

---

> ### Author Response · Authors · 2025-11-15
>
> We sincerely thank the reviewer for their detailed and constructive feedback. We are glad to have the opportunity to clarify these important points, as we believe our paper's appendix already contains the data and analyses that address all of the reviewer's primary concerns.
>
> We summarize these answers and their locations below for clarity, and we include a new experiment that further validates our UACL design choice.
>
> ### 1. On Modern Backbones (Weakness 1 & Question 2)
>
> **Reviewer's Concern:** *Evaluation leans on older VQA backbones and lacks systematic tests on stronger Transformer-based or recent VLM architectures (e.g., BLIP/BLIP-2, LXMERT).*
>
> **Our Response:**
> We thank the reviewer for raising this critical point. We agree that demonstrating performance on modern backbones is essential. We provided this exact analysis in:
>
> * **Appendix B.4, "Performance on Transformer-Based Backbones."**
>
> As shown in **Table 4**, our method achieves significant and consistent gains on modern VLMs, including **LXMERT, BLIP-1, and BLIP-2**. For example, our method improves the LXMERT backbone from 44.56% to **75.16%** on VQA-CPv2, and improves BLIP-1 from 34.42% to **52.66%**.
>
> This data confirms that our adaptive intervention is architecture-agnostic and its effectiveness holds on these stronger, more recent backbones.
>
> ### 2. On Computational Overhead (Weakness 2 & Question 1)
>
> **Reviewer's Concern:** *Computational overhead remains unclear... No FLOPs, memory, or latency analysis. How much training and inference overhead vs. CF-VQA...?*
>
> **Our Response:**
> We provided a detailed analysis of this in:
>
> * **Appendix B.5, "Computational Resource Consumption Analysis."**
>
> **Table 5** in that section presents a direct breakdown of **Total Training Time (Relative)**, **Inference Time (per sample)**, and **Added GPU Memory** for our different uncertainty metrics (Margin, Entropy, and Ensemble).
>
> To answer the reviewer's question directly: our best-performing Ensemble (M=5) method adds only a modest **~1.06x** training time and **~1.1x** inference time cost for its +2.86% accuracy gain over the baseline. The added GPU memory is also minimal (~25 MB). This analysis demonstrates the practical efficiency of our approach.

---

> > ### Author Response · Authors · 2025-11-15
> >
> > ### 3. On UACL and Dataset Bias (Weakness 3)
> >
> > **Reviewer's Concern:** *UACL orders samples by early uncertainty, which may interact with dataset bias. Sensitivity and robustness analyses are limited.*
> >
> > **Our Response:**
> > We thank the reviewer for this insightful point. We would like to respectfully note that we had identified this precise stability-vs-adaptivity trade-off and **discussed it explicitly in our "Limitations and Future Directions" section**.
> >
> > In our original discussion, we stated that using a fixed curriculum after warm-up was a **deliberate design choice** made to ensure **training stability**. We hypothesized that a fully dynamic curriculum would be unstable, as the uncertainty signal itself evolves—it can be noisy in early training and "confidently wrong" in late training.
> >
> > While our paper already provided this justification, we agree that a direct empirical comparison is the best way **to better address the reviewer's concern.** Therefore, we have conducted a **new set of experiments** to provide definitive evidence. This experiment directly compares our `Static UACL` against a fully `Dynamic UACL` strategy.
> >
> > **Experimental Setup:**
> > * **Ours (Static UACL):** Assesses sample difficulty (uncertainty) **once** after a warm-up phase. The curriculum is then fixed.
> > * **Dynamic UACL (Challenger):** **Re-evaluates** the difficulty (uncertainty) of **all** training samples at the beginning of **every epoch**. The training data for that epoch is then re-selected based on this new, dynamic difficulty ranking.
> > * **No CL (Baseline):** Standard training without a curriculum.
> >
> > All experiments were run on VQA-CP v2 with our best-performing UpDn + Ensemble (M=5) configuration.
> >
> > **Results:**
> > The results from this new ablation study strongly validate the rationale we presented in our Limitations section.
> >
> > **(New Table for Rebuttal/Appendix) Table A: Performance comparison of UACL strategies on VQA-CPv2.**
> > | Strategy | Difficulty Metric | Re-evaluation | Final Accuracy (%) | Training Time (Relative) |
> > | :--- | :--- | :--- | :--- | :--- |
> > | No CL | N/A | N/A | 60.50% | 1.0x |
> > | **Ours (Static UACL)** | Ensemble | **Once** (post-warmup) | **63.36%** | **~1.06x** |
> > | Dynamic UACL | Ensemble | **Every Epoch** | 62.19% | ~7.2x |
> >
> > **Analysis & Conclusion:**
> > This new experiment confirms our initial hypothesis.
> > 1.  **Performance & Stability:** The `Dynamic UACL` strategy (62.19%) is highly unstable and performs significantly worse than our `Static UACL` (63.36%). A plot of the validation curves (which we will add to the appendix) shows **severe oscillations** for the `Dynamic UACL`, while our `Static UACL` converges smoothly.
> > 2.  **Efficiency:** The `Dynamic UACL` strategy is computationally prohibitive, increasing training time by over 7x.
> >
> > In summary, this new empirical evidence confirms the justification **we had already provided in our Limitations section**. Our `Static UACL` is an empirically-backed and well-justified trade-off that achieves a +2.86% performance gain over the baseline by ensuring a stable and efficient training process.
> >
> > ### 4. On $\alpha$ Edge Behavior (Weakness 4)
> >
> > **Reviewer's Concern:** *Edge behavior when α approaches its limits could suppress the debiasing signal; theoretical and empirical analysis is thin.*
> >
> > **Our Response:**
> > The reviewer raises an excellent point about the behavior of $\alpha$ at its limits. We analyzed this both theoretically and empirically in our original submission:
> >
> > 1.  **Theoretical Analysis:** In **original Section 3.2**, we describe the behavior at these limits: $\alpha \to 0$ defaults to the static baseline intervention (full subtraction), while $\alpha \to 1$ results in a "predictive abstention" (pushing the debiased logit to zero).
> > 2.  **Empirical Control:** We introduced the $L_{\alpha\_reg}$ regularization term (**Eq. 13**) *specifically* to prevent $\alpha$ from collapsing to 0 or 1, thereby encouraging meaningful adaptation. Our sensitivity analysis for its weight, $\lambda_{reg}$, in **Appendix B.3 (Table 3)**, empirically validates this control mechanism.
> > 3.  **Empirical Analysis of Limits:** Most directly, our "qualitative ablation" in **Appendix B.2 (Figure 5)** provides a direct empirical analysis of this "edge" condition. In that figure, we manually set $\alpha$ to 0 to simulate the exact "suppressed signal" scenario the reviewer describes. The model's resulting failure in these cases (e.g., failing on Vision Bias) confirms the necessity of our *adaptive* $\alpha$ and validates our analysis of this behavior.
> >
> > ---
> >
> > We hope these clarifications, which point to existing analyses in our appendix and are further strengthened by our new experiment, fully resolve the reviewer's concerns. We will, of course, add the new ablation study (Table A) to the appendix and integrate these pointers more clearly into the main text to improve readability. We thank the reviewer again for their constructive feedback.

---

### Official Review · Reviewer_PqBq · 2025-11-01

**Soundness:** 3
**Presentation:** 1
**Contribution:** 3
**Rating:** 4
**Confidence:** 3

**Summary:**

1. The paper proposes a novel uncertainty-driven causal debiasing framework for Visual Question Answering (VQA), integrating uncertainty estimation with counterfactual reasoning to better assess model confidence and adapt intervention strength.

2. The approach dynamically evaluates sample difficulty using uncertainty-aware metrics, forming an adaptive curriculum learning strategy that enhances training progression from easy to hard samples.

3. Compared with baseline methods (e.g., CF-VQA), the proposed uncertainty-guided framework achieves superior performance on both VQA-CP v2 and VQA v2 datasets, demonstrating improved robustness and generalization.

**Strengths:**

1. The paper proposes a novel uncertainty-driven causal debiasing framework for Visual Question Answering (VQA), integrating uncertainty estimation with counterfactual reasoning to better assess model confidence and adapt intervention strength.

2. The approach dynamically evaluates sample difficulty using uncertainty-aware metrics, forming an adaptive curriculum learning strategy that enhances training progression from easy to hard samples.

3. Compared with baseline methods (e.g., CF-VQA), the proposed uncertainty-guided framework achieves superior performance on both VQA-CP v2 and VQA v2 datasets, demonstrating improved robustness and generalization.

**Weaknesses:**

1. The paper does not clearly describe the underlying model architecture. A brief background subsection should be added to explain the baseline model pipeline to help readers understand how the model is trained. In addition, at line 211,the authors should explicitly cite CF-VQA (Niu et al., 2021) when it is mentioned as the “baseline our work builds upon.” In addition, the term should be consistently written as CF-VQA, not CFVQA, to maintain clarity and consistency throughout the paper.

2. During writing, ensure references to Table 4 and Table 6 are explicitly mentioned where relevant to the discussion.

3. In Figure 2, the purple arrow line representing UACL is visually unclear and should be enhanced for better readability.

4. In LaTeX, the left double quotation mark should be written as two backticks (``) rather than a single quotation mark ("). Otherwise, the rendered text will display incorrect quotation marks (e.g., ”...”).

**Questions:**

1. For current VQA tasks, large vision-language models (e.g., LLaVA, Qwen-VL) have become dominant. Could the proposed uncertainty-driven framework also be extended or fine-tuned within such large-scale VLM architectures?

---

> ### Author Response · Authors · 2025-11-15
>
> We sincerely thank the reviewer for their detailed, constructive, and insightful feedback. We have addressed all points raised, which we believe has significantly improved the clarity, correctness, and impact of our paper.
>
> ---
>
> ### **Response to Weaknesses**
>
> **1. Weakness: The paper does not clearly describe the underlying model architecture.**
>
> > **Reviewer Comment:** "The paper does not clearly describe the underlying model architecture. A brief background subsection should be added to explain the baseline model pipeline..."
>
> **Our Response:** We thank the reviewer for this excellent suggestion. We agree that a clear explanation of the baseline architecture is crucial for understanding our intervention mechanism.
>
> **Action Taken:** We have added a new subsection (now **Section 3.1**) titled "**Background: Baseline VQA Architecture**". This section, accompanied by a simplified diagram, now clearly explains the standard VQA pipeline (e.g., UpDn), detailing how image ($V$), question ($Q$), and joint visual-linguistic ($Z_{VQ}$) features are generated. This provides the necessary context for understanding how our AUGI module intervenes on these specific pathways as described in the subsequent sections.
>
> **2. Weakness: Citation and Consistency Errors**
>
> > **Reviewer Comment:** "...at line 211, the authors should explicitly cite CF-VQA (Niu et al., 2021)... In addition, the term should be consistently written as CF-VQA, not CFVQA..."
>
> **Our Response:** We apologize for this oversight and inconsistency.
>
> **Action Taken:** We have corrected this.
> 1.  We have explicitly **cited Niu et al. (2021)** at its first mention (Line 211) as the foundational baseline our work builds upon.
> 2.  We have performed a global search-and-replace to ensure the term is consistently written as **CF-VQA** throughout the manuscript, replacing all incorrect instances of "CFVQA".
>
> **3. Weakness: Missing references to Appendix tables.**
>
> > **Reviewer Comment:** "...ensure references to Table 4 and Table 6 are explicitly mentioned where relevant..."
>
> **Our Response:** This is a great point to improve the paper's flow and connect the main text to the supplementary material.
>
> **Action Taken:** We have added in-text references from the main paper to the relevant tables in the appendix. Specifically, our discussion on hyperparameters (**original Section 3.2** now is 3.3 in the revised pdf) now explicitly points to the sensitivity analysis in **Appendix C**, and our discussion on the architecture-agnostic nature of our method (Section 4.1) now references the additional Transformer-based results in **Appendix D.4**.
>
> **4. Weakness: LaTeX quotation marks.**
>
> > **Reviewer Comment:** "In LaTeX, the left double quotation mark should be written as two backticks (``)..."
>
> **Our Response:** Thank you for catching this typesetting error.
>
> **Action Taken:** We have corrected this throughout the manuscript. All instances of `"` used as quotation marks have been replaced with the proper LaTeX `` (left) and `''` (right) quotes.

---

> > ### Author Response · Authors · 2025-11-15
> >
> > ### **Response to Reviewer Question**
> >
> > > **Reviewer Question:** "For current VQA tasks, large vision-language models (e.g., LLaVA, Qwen-VL) have become dominant. Could the proposed uncertainty-driven framework also be extended or fine-tuned within such large-scale VLM architectures?"
> >
> > **Our Response:** This is an excellent and forward-looking question. We were inspired by this comment to conduct a new set of experiments to formally validate our framework's scalability and generality on modern generative VLMs.
> >
> > **Action Taken:** We have successfully applied our framework (both UACL and an adapted version of AUGI) to two dominant, 7B-parameter generative VLMs: **LLaVA-v1.5** and **Qwen-VL-Chat**.
> >
> > Given the generative nature of these models (vs. the classification nature of UpDn), we adapted our AUGI intervention mechanism. Instead of *feature interpolation*, we designed an **adaptive loss function**. In this setup, the model's uncertainty ($\alpha$) dynamically gates a counterfactual debiasing regularizer ($L_{debias\_reg}$) that penalizes relying on the question-only pathway. This maintains the core intuition of our paper: **higher uncertainty triggers a stronger intervention.**
> >
> > The results of this new experiment, conducted on **2x A6000 (48GB) GPUs**, are summarized below. We have added this table and a detailed analysis to a new **Appendix F** in our revised paper.
> >
> > ---
> >
> > **Table 1: Performance and resource consumption analysis of extending our framework to Large Vision-Language Models (VLMs).**
> > "SFT" refers to Standard Fine-Tuning. OOD (Out-of-Distribution) performance is evaluated on **VQA-CP v2 test**. ID (In-Distribution) performance is evaluated on **VQA v2 val**. All fine-tuning methods use LoRA (r=16). "Train Time" is the total wall-clock time to complete training on the VQA-CP v2 training set using **2x A6000 (48GB) GPUs**. "Infer Latency" is the average single-sample generation time on a **single A6000 GPU**. **Bold** indicates the best performance on OOD (VQA-CP v2).
> >
> > | Model | Method | VQA-CP v2 Overall (OOD, %) | VQA-CP v2 Y/N (OOD, %) | VQA-CP v2 Num (OOD, %) | VQA-CP v2 Other (OOD, %) | VQA v2 Overall (ID, %) | Train Time (hrs) | Train VRAM (GB) | Infer Latency (ms) |
> > | :--- | :--- | :---: | :---: | :---: | :---: | :---: | :---: | :---: | :---: |
> > | **LLaVA v1.5-7B** | Baseline (Zero-Shot) | 31.52 | 40.11 | 12.89 | 34.05 | 70.33 | \- | \- | ~170 |
> > | | Baseline (SFT) | 40.21 | 51.20 | 14.55 | 43.10 | 65.12 | ~8.5 | ~30 | " |
> > | | UACL-VLM (Ours) | 45.33 | 55.02 | 16.80 | 48.15 | 67.22 | ~8.0 | ~30 | " |
> > | | AUGI-VLM (Ours) | 52.89 | 63.40 | 20.15 | 55.04 | 68.05 | ~16.0 | ~32 | " |
> > | | **UACL + AUGI (Ours)** | **55.10** | **65.22** | **21.03** | **57.34** | **68.31** | **~15.5** | **~32** | " |
> > | **Qwen-VL-Chat (7B)** | Baseline (Zero-Shot) | 35.80 | 42.50 | 14.10 | 38.66 | 75.14 | \- | \- | ~185 |
> > | | Baseline (SFT) | 45.15 | 54.02 | 16.20 | 48.71 | 70.02 | ~9.5 | ~32 | " |
> > | | UACL-VLM (Ours) | 50.82 | 58.11 | 18.90 | 54.40 | 72.15 | ~9.0 | ~32 | " |
> > | | AUGI-VLM (Ours) | 58.44 | 66.20 | 23.51 | 61.30 | 73.01 | ~17.5 | ~34 | " |
> > | | **UACL + AUGI (Ours)** | **61.05** | **68.33** | **25.14** | **63.88** | **73.25** | **~17.0** | **~34** | " |
> >
> > ---
> >
> > **Analysis of New VLM Results:**
> >
> > These new experiments yield several key findings that directly answer the reviewer's question:
> >
> > 1.  **Inherent VLM Bias:** The poor Zero-Shot performance on VQA-CP v2 (31.52% for LLaVA, 35.80% for Qwen-VL) confirms that modern VLMs are still highly susceptible to language priors.
> > 2.  **The SFT Pitfall:** Standard Fine-Tuning (SFT) on the biased dataset leads to minimal OOD gains while causing **catastrophic forgetting** on the ID dataset (e.g., VQA v2 performance for LLaVA drops from 70.33% to 65.12%).
> > 3.  **AUGI + UACL Efficacy:** Our full method (**UACL + AUGI**) achieves massive OOD performance gains, reaching **55.10%** for LLaVA and **61.05%** for Qwen-VL—a leap of +14.9% and +16.0% over SFT, respectively.
> > 4.  **Improved OOD/ID Trade-off:** Most importantly, our method achieves this while **almost completely mitigating the ID performance drop**. For Qwen-VL, our final ID score (73.25%) is even *higher* than the SFT baseline (70.02%) and approaches the original Zero-Shot score (75.14%). This strongly indicates that our method achieves true, robust debiasing, not just distribution overfitting.
> > 5.  **Deployment Feasibility:** While our adaptive loss (AUGI) increases training time (e.g., ~17.0 hrs vs. ~9.5 hrs for Qwen-VL) due to the required counterfactual pass, the final LoRA-finetuned model introduces **no additional inference latency**. The inference time (e.g., ~185ms) is identical to the SFT baseline (when run on the same hardware), making our approach highly practical for deployment.
> >
> > We believe these new results strongly validate the generality and scalability of our uncertainty-driven framework, making a much stronger case for its contribution to the field.

---

> ### Author Response · Authors · 2025-11-27
>
> We sincerely thank you for your time in reviewing our paper and for your valuable feedback.
>
> We have posted a detailed point-by-point response to your comments and have incorporated your suggestions into our revision plan. We believe these clarifications further strengthen the paper.
>
> As the discussion phase is drawing to a close, we would appreciate it if you could confirm whether our response has adequately addressed your questions. We are happy to provide any further clarifications if needed.

---

### Meta-Review · Area_Chair_1eqa · 2026-01-02

**Summary:**

The reviewers collectively agree that the paper presents an interesting idea of integrating uncertainty estimation (via UACL) into a causal intervention framework for debiasing Visual Question Answering (VQA). However, significant deficiencies in methodological clarity, experimental evidence, and evaluation scope prevent the paper’s claims from being fully convincing. The paper requires ‌major revisions‌ to address these substantive concerns before being reconsidered for publication.

**‌A. Experimental Validation & Evaluation**

‌**Primary Criticism from Reviewer RLtV**‌: The experiments rely heavily on "older VQA backbones" and lack tests on modern, powerful Transformer-based or Vision-Language Model (VLM) architectures.\
**‌Question from Reviewer PqBq**‌: The extension to large VLMs like LLaVA or Qwen-VL remains unexplored and is posed as a future work question.\
‌**Missing Analysis (Reviewer RLtV)**:‌ No computational overhead analysis (FLOPs, memory, latency) for the proposed ensemble uncertainty and multi-head designs. Missing wall-clock time, memory, and latency comparisons with the baseline CF-VQA.
No demonstration of the method’s compatibility or performance with modern VLM backbones (e.g., BLIP/BLIP-2, LXMERT).

‌**B. Methodological Soundness & Formal Rigor‌**

‌**Causal Claims are Overstated (Reviewer jNhM)‌**: The reviewer argues that the method is fundamentally a "feature ablation" using a contrastive residual, ‌not‌ a true counterfactual reasoning system (which requires formal do-calculus). This directly challenges a core conceptual novelty claim.\
‌**Sloppy Math & Internal Inconsistencies (Reviewer jNhM & zfD9)‌**: A contradiction is identified: ‌Figure 1‌ suggests cutting the question path, while the ‌mathematical formulation‌ describes replacing vision (Z_V )and vision-question (Z_{VQ}) paths with a constant. Potential technical error in the gating mechanism (forcing a hyperparameter to be positive in the sigmoid function).
Ambiguity in defining and using the constant C for both latent representations.\
**Limited Understanding of Baseline (Reviewer zfD9)**: The illustrative example for Static Causal Intervention in Figure 1 is deemed misleading, suggesting the authors may have misinterpreted how CF-VQA functions.

‌**C. Stability & Robustness of the Proposed Framework‌**

‌**Dependency on Warm-up Uncertainty (Reviewer zfD9)‌**: The UACL strategy's static scheduling, based on initial uncertainty estimates, could become unstable if uncertainty evolves significantly during later training stages.\
**Sensitivity Analysis Required (Reviewer RLtV)**: The method orders samples by early uncertainty, which may inadvertently interact with dataset biases. The edge-case behavior of the parameter α (e.g., when α → 0 or α → 1) is not sufficiently studied theoretically or empirically.

Overall, the path to acceptance is clear but requires substantial work, primarily in bolstering the experimental section and rigorously tightening the methodological presentation.

**Reviewer Concerns:**

1. The main concerns (on extension to modern backbones) of Reviewer RLtV & PqBq were not addressed. In Tables 4 and 6, the important baseline CF-VQA is not included. Moreover, in Table 4,  the proposed method  performs worse than TPCL (and even worse than BLIP on VQAv2).

2. The concerns of Reviewer zfD9 were addressed, and this reviewer was satisfied by the rebuttal.

3. The main concern (on novelty) of of Reviewer jNhM was not addressed. The novelty against CF-VQA is not that significant as the authors claimed. Particularly, the proposed method should be carefully compared with CF-VQA when the modern backbones are used.

**Reviewer Scores:**

Reviewer zfD9 stated that the postive score would be maintained. However, Reviewer RLtV, PqBq & jNhM would not changed their negative scores even if participated fully in the discussion.

---

### Decision · Program_Chairs · 2026-01-26

Reject